# Self-Weighted Contrastive Learning among Multiple Views for Mitigating Representation Degeneration

**Jie Xu**[1]    **Shuo Chen**[2]    **Yazhou Ren**[1]    **Xiaoshuang Shi**[1]
**Heng Tao Shen**[1]    **Gang Niu**[2]    **Xiaofeng Zhu**[1,*]
[1]University of Electronic Science and Technology of China, China
[2]RIKEN Center for Advanced Intelligence Project, Japan

## Abstract

Recently, numerous studies have demonstrated the effectiveness of *contrastive learning* (CL), which learns feature representations by pulling in positive samples while pushing away negative samples. Many successes of CL lie in that there exists semantic consistency between data augmentations of the same instance. In *multi-view scenarios*, however, CL might cause *representation degeneration* when the collected multiple views inherently have inconsistent semantic information or their representations subsequently do not capture sufficient discriminative information. To address this issue, we propose a novel framework called *SEM: SElf-weighted Multi-view contrastive learning with reconstruction regularization*. Specifically, SEM is a general framework where we propose to first measure the discrepancy between pairwise representations and then minimize the corresponding self-weighted contrastive loss, and thus making SEM adaptively strengthen the useful pairwise views and also weaken the unreliable pairwise views. Meanwhile, we impose a self-supervised reconstruction term to regularize the hidden features of encoders, to assist CL in accessing sufficient discriminative information of data. Experiments on public multi-view datasets verified that SEM can mitigate representation degeneration in existing CL methods and help them achieve significant performance improvements. Ablation studies also demonstrated the effectiveness of SEM with different options of weighting strategies and reconstruction terms.

## 1 Introduction

*Contrastive learning* (CL) explicitly enlarges the feature representation similarity between semantic-relevant samples, and it is adept at capturing high-level semantics while discarding irrelevant information. This learning paradigm has facilitated many research and application fields, such as visual representation [1, 2], text understanding [3, 4], and cross-modal agreement [5, 6, 7]. Samples with consistent semantics are typically constructed as positive sample pairs for CL loss (*e.g.*, InfoNCE [8]), which motivates multi-view learning scenarios [9, 10] where researchers focus on exploring common semantics among multi-view data. However, this kind of data usually is with heterogeneous views and thus cannot be directly processed by previous CL methods with two shared network branches.

To handle this situation, many *multi-view contrastive learning* (MCL) methods [11, 12, 13, 14] have been proposed, which treats multiple views as positive sample pairs and achieves important progresses in exploring multi-view common semantics (see Sec. 2 for details). Nevertheless, we find that CL might cause *representation degeneration* that the representations of high-quality views tend to degenerate. This may make the MCL methods perform worse than the optimal single view (see Sec. 3.1 and Sec. 4.1), and thus heavily limiting the usability of MCL in practical scenarios. Although

---

[*]Corresponding Author (seanzhuxf@gmail.com). Code link: *https://github.com/SubmissionsIn/SEM*.

37th Conference on Neural Information Processing Systems (NeurIPS 2023).

several CL work [15, 16] proposed different CL losses aiming at increasing robustness to noise and made important advances on vision and graph data, our experiments discover that these CL losses are still fragile in multi-view scenarios as multi-view data are with more diversity than single-view data. Different from changing CL loss, recent MCL methods [14, 17] focused on changing model structures and successfully improved the effectiveness of clustering the learned representations. Nevertheless, representation degeneration still exists in many cases and it requires further solutions.

We find that there could be two reasons leading to representation degeneration in MCL. **I)** The quality difference among multiple views. The success of CL is based on the priori condition that the constructed positive sample pair has semantic consistency, which generally holds in previous CL applications [1, 5, 8]. Unfortunately, for multi-view learning, the collected views usually have quality difference and the semantic of positive sample pairs might be inconsistent due to view diversity. Consequently, CL causes the representation degeneration of high-quality views due to the existence of low-quality views. **II)** Losing discriminative information during data processing. Multi-view data typically involve heterogeneous data forms [9, 18], *e.g.*, different dimensions, modalities, and sparsity. For achieving MCL, the model needs to transform heterogeneous multi-view data into the same form with different encoders. However, data transformation could lose discriminative information as this process has no supervised signals for maintaining information. As a result, CL might miss multiple views' common semantics and focus on semantic-irrelevant information due to inductive bias.

To this end, we propose *SElf-weighted Multi-view contrastive learning with reconstruction regularization (SEM)* as shown in Figure 1 that takes the $m, n, o$-th views in $V$ views as an example (where $\mathcal{W}^{m,n}$ denotes the pairwise weight, $\mathcal{L}_{CL}^{m,n}$ is the contrastive loss, and $\mathbf{Z}^m$ is the learned representations). Specifically, SEM minimizes self-weighted contrastive losses $\mathcal{W}^{m,n}\mathcal{L}_{CL}^{m,n}$ and $\mathcal{W}^{n,o}\mathcal{L}_{CL}^{n,o}$ after measuring the discrepancy between pairwise views' representations, *i.e.*, $(\mathbf{Z}^m, \mathbf{Z}^n)$ and $(\mathbf{Z}^n, \mathbf{Z}^o)$, respectively. This makes SEM adaptively strengthen CL between the useful pairwise views and also weaken CL between the unreliable pairwise views. Meanwhile, SEM takes self-supervised reconstruction objectives as regularization terms ($\mathcal{R}^m$, $\mathcal{R}^n$, and $\mathcal{R}^o$) on the hidden features ($\mathbf{H}^m$, $\mathbf{H}^n$, and $\mathbf{H}^o$) of encoders for individual views, respectively. This reconstruction regularization assists CL in accessing sufficient discriminative information hidden in raw input data ($\mathbf{X}^m$, $\mathbf{X}^n$, and $\mathbf{X}^o$), which could be implemented by exist-

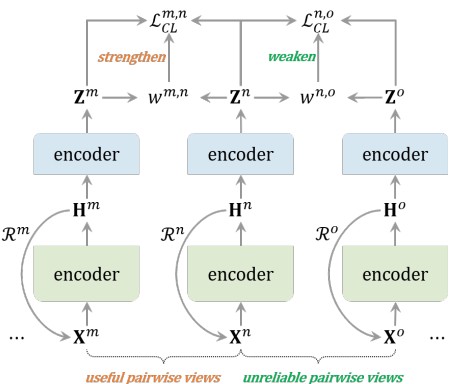

Figure 1: The framework of SEM. It leverages different networks to extract information of different views and conducts the proposed self-weighted multi-view contrastive learning with reconstruction regularization.

ing information encoder-decoder models, *e.g.*, AE [19], DAE [20], and MAE [21]. In SEM, the representations and pairwise weights are alternatively updated to mutually enhance one another.

In summary, our contributions are: **I)** We propose a novel general framework SEM that leverages self-weighting and information reconstruction to address representation degeneration in MCL. **II)** We provide three options with different advantages to implement the weighting strategy of SEM including class mutual information, JS divergence, and maximum mean discrepancy. **III)** Theoretical and experimental analysis verified the effectiveness of SEM. It helps many CL methods (*e.g.*, InfoNCE [8], RINCE [15], and PSCL [16]) achieve significant performance improvements in multi-view scenarios.

## 2 Related Work

**Contrastive learning (CL)** As a popular self-supervised learning paradigm, CL focuses on learning semantically informative representations for downstream tasks [22, 23, 24, 25]. The most widely used loss function is InfoNCE [8] which pulls in the representations between positive sample pairs while pushing away that between negative sample pairs. Some work have attempted to explain the reasons for the success of applying InfoNCE, *e.g.*, from perspectives of mutual information [8, 26], task-dependent view [27], or deep metric learning [28, 29]. Furthermore, [30, 31] pointed out to conduct CL with reconstruction regularization to achieve robust representations for downstream tasks. RINCE [15] (a short name of Robust InfoNCE) is a variant of InfoNCE contrastive loss that considers noise in false positive sample pairs. The recent work [16] investigates CL without

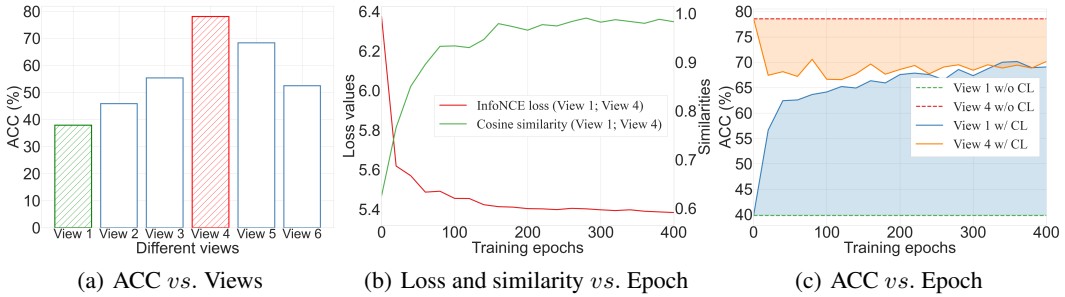

Figure 2: (a) Clustering accuracy of individual views on Caltech dataset. (b) Contrastive loss and representation similarity between view 1 and view 4. (c) Clustering accuracy of view 1 and view 4 during contrastive learning.

conditional independence assumption on positive sample pairs and proposes a population spectral contrastive loss (we call it PSCL for short). Despite important progresses have been made, in this work, we discover that these CL losses are still fragile in multi-view scenarios where data qualities are hard to be guaranteed, and even the reconstruction regularized CL is not enough.

**Multi-view contrastive learning (MCL)** Different from many CL methods that usually generate two inputs by data augmentation [32], MCL aims to handle multi-view data widely exiting in real-world applications. Multi-view data often contain more than two views/modalities and they naturally form multiple inputs [33, 34, 35]. Since the semantic consistency among multiple views is not guaranteed, it is challenging to capture the useful information in multi-view data, while considering the side effects of harmful information. Therefore, MCL attracts increasing attention in recent years [36, 37, 38]. For example, CMC [11] empirically shows that MCL performed with more scene views obtains the better representations with semantic information. DCP [39] leverages the maximization of mutual information to conduct consistency learning across different views and aims to achieve a provable sufficient and minimal representation. MFLVC [14] observes the conflict between consistency and reconstruction objectives in encoder-decoder frameworks and proposes to learn multi-level features for multiple views. DSIMVC [17] establishes a theoretical framework to reduce the risk of clustering performance degradation from semantic inconsistent views. Although satisfactory results are achieved in many cases, the representation degeneration caused by CL is still not well considered and addressed. In this paper, we point out that the representation degeneration could seriously limit the application of CL in multi-view scenarios, and propose the discrepancy-based self-weighted MCL to address it.

**Notations** This paper leverages bold uppercase characters and bold lowercase characters to denote matrices and vectors, respectively. Operator $\|\cdot\|_2$ denotes vector $\ell_2$-norm and operator $\|\cdot\|_F$ is matrix $F$-norm. $\{\mathbf{x}_i^v \in \mathbf{X}^v\}_{i=1,2,\ldots,N}^{v=1,2,\ldots,V}$ denotes the multi-view dataset with $N$ samples in $V$ views.

## 3 Methodology

This section first illustrates the phenomenon of representation degeneration in multi-view contrastive learning. To address this issue, we then establish a general framework of *SEM: SElf-weighed Multi-view contrastive learning with reconstruction regularization*. To implement the SEM framework, we further provide different options of weighing strategy, contrastive loss, and reconstruction term.

### 3.1 Motivation: Representation Degeneration in Multi-View Contrastive Learning

Researchers proposed many contrastive learning approaches and also achieved plenty of progress in multi-view learning. However, multi-view contrastive learning might result in the representation degeneration of high-quality views (*i.e.*, those views contain rich semantic information) due to the diversity of multi-view data. Specifically, we illustrate it in Figure 2 that takes a popular multi-view dataset Caltech [40] (6 views) as an example. We leverage unsupervised linear clustering accuracy obtained by K-Means [41] to evaluate the representation quality of containing class-level semantics.

Firstly, we leverage self-supervised autoencoders (the setting is shown in Appendix B) to pretrain the representations of each view's data. In Figure 2(a), one can find that different views inherently have different levels of discriminative information and exhibit different qualities, where the worst (view 1)

and the best (view 4) have a large gap. Then, we adopt InfoNCE loss to perform contrastive learning between view 1 and view 4 in Figure 2(b), and record the clustering accuracy of their representations in Figure 2(c). We can observe that InfoNCE loss is well-minimized, which makes the representation similarity (evaluated by cosine) between view 1 and view 4 converge to 1.0. The performance on view 1 gradually increases. Nevertheless, the cost is that the representations of view 4 degenerate, on which the useful discriminative information reduces and thus the performance gradually decreases.

In multi-view learning, quality difference among multiple views is a common phenomenon. However, the representation degeneration in multi-view contrastive learning might make the representations of some high-quality views tend to be mediocre and thus miss their useful discriminative information.

## 3.2 Self-Weighted Multi-View Contrastive Learning with Reconstruction Regularization

To mitigate representation degeneration in multi-view contrastive learning, we propose a simple but effective framework called *SEM: SElf-weighted Multi-view contrastive learning with reconstruction regularization* as shown in Figure 1. Specifically, given view-specific data $\mathbf{X}^v \in \mathbb{R}^{N \times d_v}$, we let $\mathbf{Z}^v \in \mathbb{R}^{N \times z}$ denote the corresponding new representations learned by a view-specific encoder. Between $\mathbf{X}^v$ and $\mathbf{Z}^v$, we record a precursor state of representations as $\mathbf{H}^v \in \mathbb{R}^{N \times h_v}$ (termed as hidden features), and the encoder is partitioned into two parts (the front and back parts are stacked and denoted as $f^v$ and $g^v$ sequentially). For the $v$-th view, we let $\Psi^v$ and $\Phi^v$ denote the network parameters of $f^v$ and $g^v$, respectively, and then the view-specific model can be formulated as follows:

$$\mathbf{Z}^v = g^v(\mathbf{H}^v; \Phi^v) = g^v(f^v(\mathbf{X}^v; \Psi^v); \Phi^v). \tag{1}$$

In SEM, we leverage $\mathcal{L}_{CL}^{m,n}(\mathbf{Z}^m, \mathbf{Z}^n)$ to denote a contrastive loss[2], and let $\lambda > 0$ denote a trade-off coefficient on regularization terms. Then, SEM is trained by minimizing the following objective:

$$\sum_{m,n} \mathcal{W}^{m,n} \mathcal{L}_{CL}^{m,n}(\mathbf{Z}^m, \mathbf{Z}^n) + \lambda \sum_v \mathcal{R}^v(\mathbf{X}^v, \mathbf{H}^v), \tag{2}$$

where $\mathcal{W}^{m,n}$ is the pairwise weight between the $m$-th and the $n$-th views, and $\mathcal{R}^v(\mathbf{X}^v, \mathbf{H}^v)$ denotes the reconstruction regularization on $\mathbf{H}^v$. We define $\mathcal{D}(\mathbf{Z}^m, \mathbf{Z}^n)$ as the discrepancy between $\mathbf{Z}^m$ and $\mathbf{Z}^n$ and denote $\mathcal{F}$ as a negative correlation function. Then, in SEM, the pairwise weight is updated by

$$\mathcal{W}^{m,n} = \mathcal{F}(\mathcal{D}(\mathbf{Z}^m, \mathbf{Z}^n)). \tag{3}$$

**Self-weighting** In unsupervised settings, it is hard to know which representations within $\{\mathbf{Z}^v\}_{v=1}^V$ contain useful semantic information and which are with more noise. To mitigate the representation degeneration caused by contrastive learning, SEM needs to be adaptive to quality difference among multiple views. Therefore, different from using equal-sum manner [11, 14, 17] (*e.g.*, $\sum_{m,n} \mathcal{L}_{CL}^{m,n}$), we propose to use the pairwise weighted multi-view contrastive loss, *i.e.*, $\sum_{m,n} \mathcal{W}^{m,n} \mathcal{L}_{CL}^{m,n}$. Here, $\mathcal{W}^{m,n}$ leverages the discrepancy to achieve the adaptive self-weighting. Concretely, if two views are useful pairwise views and both with informative semantics, contrastive learning between them is adaptively strengthened; if two views are unreliable pairwise views (for example, one or two of them are with less informative semantics), contrastive learning between them is adaptively weakened.

**Reconstruction regularization** In Eq. (2), $\mathcal{R}^v(\mathbf{X}^v, \mathbf{H}^v)$ acts as a self-supervised objective to transfer as much discriminative information as possible from $\mathbf{X}^v$ to $\mathbf{H}^v$. When we record $\mathbf{H}^v$ as the hidden features in encoder networks, the information transfer path can be described as $\mathbf{X}^v \to \mathbf{H}^v \to \mathbf{Z}^v, v \in \{1, 2, \ldots, V\}$. However, information losing might occur in the processing of $\mathbf{X}^v \to \mathbf{H}^v$ such that discriminative information from some views' data is lost, and thus making contrastive learning among $\{\mathbf{Z}^v\}_{v=1}^V$ focus on harmful noise instead of common semantics across multiple views. To this end, on hidden features $\mathbf{H}^v$, our SEM leverages $\mathbf{X}^v$ to build the reconstruction regularization $\mathcal{R}^v(\mathbf{X}^v, \mathbf{H}^v)$ to assist contrastive learning in accessing sufficient discriminative information from raw data.

---

[2]$\mathcal{L}_{CL}^{m,n}(\mathbf{Z}^m, \mathbf{Z}^n)$ can be easily replaced by previous contrastive losses, *e.g.*, InfoNCE [8], RINCE [15], and PSCL [16]. Let $\mathcal{P}$ denote the set of positive sample pairs and $\mathcal{N}$ is the set of negative sample pairs in the $m, n$-th views, $q$ and $\alpha$ are hyper-parameters of RINCE, then the three contrastive losses could be formulated as follows:

$$\mathcal{L}_{InfoNCE}^{m,n} = -\mathbb{E}_{s^+ \in \mathcal{P}} \left[ s^+ - \log \left( e^{s^+} + \sum_{s^- \in \mathcal{N}} e^{s^-} \right) \right],$$

$$\mathcal{L}_{RINCE}^{m,n} = -\mathbb{E}_{s^+ \in \mathcal{P}} \left[ \frac{1}{q} \cdot e^{q \cdot s^+} - \frac{1}{q} \cdot \left( \alpha \cdot \left( e^{s^+} + \sum_{s^- \in \mathcal{N}} e^{s^-} \right) \right)^q \right],$$

$$\mathcal{L}_{PSCL}^{m,n} = -\mathbb{E}_{s^+ \in \mathcal{P}} \left[ 2 \cdot s^+ \right] + \mathbb{E}_{s^- \in \mathcal{N}} \left[ (s^-)^2 \right],$$

where $s^+$ $(s^-)$ denotes the cosine distance between the representations of positive (negative) sample pair.

### 3.3 Different Options for Implementing the SEM Framework

The crucial components of our proposed SEM as Eq. (2) include the weighting strategy $\mathcal{W}^{m,n}$, contrastive loss $\mathcal{L}_{CL}^{m,n}$, and regularization term $\mathcal{R}^v$. Next, we concentrate on the implementations of $\mathcal{W}^{m,n}$ (including JSD, MMD, and CMI) and briefly introduce the implementations of $\mathcal{L}_{CL}^{m,n}$ and $\mathcal{R}^v$.

**Discrepancy measurements of weighting strategy** When implementing $\mathcal{W}^{m,n} = \mathcal{F}(\mathcal{D}(\mathbf{Z}^m, \mathbf{Z}^n))$ in Eq. (3), many methods can measure the discrepancy $\mathcal{D}(\mathbf{Z}^m, \mathbf{Z}^n)$. Firstly, we can transfer representations to a probability distribution and leverage Jensen-Shannon divergence (JSD) to compute the discrepancy $\mathcal{D}_{JSD}(\mathbf{Z}^m, \mathbf{Z}^n)$. The advantages of JSD are its symmetry and simplicity, but it might be inapplicable when two distributions are non-overlapping. Furthermore, we can leverage maximum mean discrepancy (MMD) as the second method to obtain the discrepancy $\mathcal{D}_{MMD}(\mathbf{Z}^m, \mathbf{Z}^n)$. MMD can effectively measure non-overlapping two distributions, but it has higher complexity than JSD[3].

Actually, both JSD and MMD leverage all information of representations $\mathbf{Z}^m$ and $\mathbf{Z}^n$. However, the semantic-irrelevant information or random noise might also be embedded in $\mathbf{Z}^m$ and $\mathbf{Z}^n$. Moreover, what we expect to obtain is the mutual relation of their most representative semantic information. To this end, we propose Class Mutual Information (CMI) as the third method to obtain the discrepancy $\mathcal{D}_{CMI}(\mathbf{Z}^m, \mathbf{Z}^n)$. To be specific, since it is difficult to accurately estimate the mutual information (denoted as $I$) for multi-dimensional continuous variables $I(\mathbf{Z}^m; \mathbf{Z}^n)$, we denote $\mathbf{y}^m$ and $\mathbf{y}^n$ as 1-dimensional discrete vectors and change estimating $I(\mathbf{Z}^m; \mathbf{Z}^n)$ to computing $I(\mathbf{y}^m; \mathbf{y}^n)$ such that:

$$\mathcal{W}^{m,n} \approx \mathcal{F}(1/I(\mathbf{Z}^m; \mathbf{Z}^n)) \approx \mathcal{F}(1/I(\mathbf{y}^m; \mathbf{y}^n)), s.t. \underset{\mathbf{y}^m, \mathbf{y}^n}{\arg\max} I(\mathbf{Z}^m; \mathbf{y}^m) + I(\mathbf{Z}^n; \mathbf{y}^n). \quad (4)$$

Intuitively, discrete class information in $\mathbf{Z}^v$ ($v \in \{m, n\}$) is 1-dimensional as well as the most representative information. Hence, we can optimize K-Means objective to extract the class information:

$$\mathbf{Y}^{v*} = \underset{\mathbf{Y}^v, \mathbf{C}^v}{\arg\max} \|\mathbf{Z}^v - \mathbf{Y}^v \mathbf{C}^v\|_F^2, s.t. \mathbf{Y}^v (\mathbf{Y}^v)^T = \mathbf{I}_N, \mathbf{Y}^v \in \{0, 1\}^{N \times K}, \quad (5)$$

where $\mathbf{C}^v \in \mathbb{R}^{K \times z}$ denotes the $K$ cluster centers of $\mathbf{Z}^v$. $\mathbf{Y}^{v*} \in \{0, 1\}^{N \times K}$ is the indicator matrix that can be further transformed to 1-dimensional discrete vector $\mathbf{y}^v$ by defining $y_i^v := \arg\max_j y_{ij}^{v*}$ where $y_i^v \in \mathbf{y}^v, y_{ij}^{v*} \in \mathbf{Y}^{v*}$. In this way, the class information in $\mathbf{Z}^m$ and $\mathbf{Z}^n$ can be compressed into $\mathbf{y}^m$ and $\mathbf{y}^n$, respectively. Then, the class mutual information $I(\mathbf{y}^m; \mathbf{y}^n)$ is normalized and the discrepancy measurement $\mathcal{D}_{CMI}(\mathbf{Z}^m, \mathbf{Z}^n)$ between pairwise views is defined as follows:

$$\mathcal{D}_{CMI}(\mathbf{Z}^m, \mathbf{Z}^n) = \frac{H(\mathbf{y}^m) + H(\mathbf{y}^n)}{2 \cdot I(\mathbf{y}^m; \mathbf{y}^n)}, \quad (6)$$

where $H(\mathbf{y}^m) = -\sum_{i=1}^N p(y_i^m) \log p(y_i^m)$ is the cross-entropy of $\mathbf{y}^m$. This design of CMI has at least two advantages: 1) It is conducive to maintaining the representative class information while filtering out noise information; 2) Calculation is easy and owns better physical meaning.

Finally, it is also flexible to implement the negative correlation function $\mathcal{F}$. Considering $\mathcal{W}^{m,n} \geq 0$, we base on the three different discrepancies and simply give the following weighting strategies:

$$\begin{aligned}
\mathcal{W}_{CMI}^{m,n} &= \mathcal{F}_{CMI}(\mathcal{D}_{CMI}(\mathbf{Z}^m, \mathbf{Z}^n)) = e^{1/\mathcal{D}_{CMI}(\mathbf{Z}^m, \mathbf{Z}^n)} - 1, \\
\mathcal{W}_{JSD}^{m,n} &= \mathcal{F}_{JSD}(\mathcal{D}_{JSD}(\mathbf{Z}^m, \mathbf{Z}^n)) = e^{1-\mathcal{D}_{JSD}(\mathbf{Z}^m, \mathbf{Z}^n)} - 1, \\
\mathcal{W}_{MMD}^{m,n} &= \mathcal{F}_{MMD}(\mathcal{D}_{MMD}(\mathbf{Z}^m, \mathbf{Z}^n)) = e^{-\mathcal{D}_{MMD}(\mathbf{Z}^m, \mathbf{Z}^n)}.
\end{aligned} \quad (7)$$

**Compatibility for contrastive learning** When implementing the contrastive loss $\mathcal{L}_{CL}^{m,n}$, it should be pointed out that multi-view contrastive learning usually has to handle more than two views (*i.e.*,

---

[3]We write $\hat{\mathbf{z}}_i^m \in \hat{\mathbf{Z}}^m = Softmax(\mathbf{Z}^m)$. $k(\mathbf{z}_i, \mathbf{z}_j)$ denotes the inner product of $\phi(\mathbf{z}_i)$ and $\phi(\mathbf{z}_j)$, where $\phi(\cdot)$ denotes the mapping (*e.g.*, by Gaussian kernel) to project representations into Reproducing Kernel Hilbert Space (RKHS). Then, $\mathcal{D}_{JSD}(\mathbf{Z}^m, \mathbf{Z}^n)$ and $\mathcal{D}_{MMD}(\mathbf{Z}^m, \mathbf{Z}^n)$ can be formulated as follows:

$$\mathcal{D}_{JSD}(\mathbf{Z}^m, \mathbf{Z}^n) = \frac{1}{2} \sum_{i=1}^N p(\hat{\mathbf{z}}_i^m) \log \left( \frac{2 \cdot p(\hat{\mathbf{z}}_i^m)}{p(\hat{\mathbf{z}}_i^m) + p(\hat{\mathbf{z}}_i^n)} \right) + \frac{1}{2} \sum_{i=1}^N p(\hat{\mathbf{z}}_i^n) \log \left( \frac{2 \cdot p(\hat{\mathbf{z}}_i^n)}{p(\hat{\mathbf{z}}_i^n) + p(\hat{\mathbf{z}}_i^m)} \right),$$

$$\mathcal{D}_{MMD}(\mathbf{Z}^m, \mathbf{Z}^n) = \frac{1}{N^2} \left[ \sum_{i=1}^N \sum_{j=1}^N k(\mathbf{z}_i^m, \mathbf{z}_j^m) + \sum_{i=1}^N \sum_{j=1}^N k(\mathbf{z}_i^n, \mathbf{z}_j^n) - 2 \sum_{i=1}^N \sum_{j=1}^N k(\mathbf{z}_i^m, \mathbf{z}_j^n) \right].$$

$\{\mathbf{Z}^v\}_{v=1}^V, V > 2$), which is different from two-view setting (*e.g.*, $\{\mathbf{Z}^1, \mathbf{Z}^2\}$) in traditional contrastive learning. To make our SEM framework be compatible with previous contrastive learning methods, we construct positive/negative sample pairs as follows. Specifically, for two views $\{\mathbf{z}_i^m \in \mathbf{Z}^m, \mathbf{z}_j^n \in \mathbf{Z}^n\}$, the positive sample pairs are $\{\mathbf{z}_i^m, \mathbf{z}_i^n\}_{i=1,\dots,N}$; for any $\mathbf{z}_i^m$, its negative sample pairs are $\{\mathbf{z}_i^m, \mathbf{z}_j^v\}_{j \neq i}^{v=m,n}$. Cosine with a temperature parameter $\tau$ is leveraged to measure the representation distance between pairs, *i.e.*, $s = 1/\tau \cdot \langle \mathbf{z}_i^m, \mathbf{z}_j^n \rangle / \|\mathbf{z}_i^m\|_2 \|\mathbf{z}_j^n\|_2$. Then, we compute the contrastive loss between two views and sum all combinations as Eq. (2). We formulated three contrastive losses in Sec. 3.2, and the experiments in Sec. 4.1 will verify the compatibility of our SEM framework to them.

**Reconstruction regularization** When implementing the regularization term $\mathcal{R}^v(\mathbf{X}^v, \mathbf{H}^v)$ in Eq. (2), we are motivated by the information encoding-decoding process [14, 19, 30], and stack a view-specific decoder $f_-^v$ with network parameter $\Omega^v$ on each view's $\mathbf{H}^v$ to perform data recovery of $\mathbf{X}^v$. In this way, the regularization term in SEM can be implemented with the reconstruction loss of autoencoders[4], whose encoder-decoder models can make hidden features preserve discriminative information of data. When decoder generally rebuilds $\mathbf{X}^v$ with $\mathbf{H}^v$, we can believe that $\mathbf{H}^v$ compresses the sufficient information of $\mathbf{X}^v$, for promoting contrastive learning fully access discriminative information of data.

---

**Algorithm 1:** Self-weighted multi-view contrastive learning with reconstruction regularization

---

**Input:** Dataset $\{\mathbf{X}^v\}_{v=1}^V$, Training epochs $E$, Step size $S$, Batch size $n$, Hyper-parameter $\lambda$
Initialize $\{\Psi^v, \Omega^v\}_{v=1}^V$ by Eq. (8) and initialize $\{\mathcal{W}^{m,n}\}_{m,n=1}^V$ with $\{\mathbf{H}^v\}_{v=1}^V$ like Eq. (7)
**for** $e \in \{1, 2, \dots, E\}$ **do**
    **for** $b \in \{1, 2, \dots, N/n\}$ **do**
        Pick mini-batch data $\{\{\mathbf{x}_i^v\}_{i=(b-1)n+1}^{bn}\}_{v=1}^V$ from $\{\mathbf{X}^v\}_{v=1}^V$
        Compute the gradient of loss via Eq. (2) on the mini-batch data
        Update $\{\Phi^v, \Psi^v, \Omega^v\}_{v=1}^V$ via Adam [42] optimizer
    **if** $\mathrm{mod}(e, S) == 0$ **then**
        Update $\{\mathcal{W}^{m,n}\}_{m,n=1}^V$ with $\{\mathbf{Z}^v\}_{v=1}^V$ by Eq. (7)
**Output:** Model parameters $\{\Phi^v, \Psi^v\}_{v=1}^V$

---

The training steps of SEM is summarized in Algorithm 1, where representations and weights are updated alternatively to make them promote each other. $E$ denotes total training epochs, and the step size $S$ denotes the number of training epochs after each update of pairwise weights. As we cannot obtain meaningful $\{\mathbf{y}^v\}_{v=1}^V$ before we start training neural networks, we first obtain meaningful $\{\mathbf{H}^v\}_{v=1}^V$ by pre-training the model with Eq. (8), and then initialize $\{\mathcal{W}^{m,n}\}_{m,n=1}^V$ with $\{\mathbf{H}^v\}_{v=1}^V$.

### 3.4 Theoretical Analysis

In this part, we theoretically analyze the mechanism of SEM in exploring mutual information among multiple views while mitigating representation degeneration. All proofs are given in Appendix A.

Considering SEM with InfoNCE loss and CMI weighting strategy, we have the following theorem indicating that minimizing the self-weighted contrastive loss keeps maximizing the mutual information between useful pairwise views, as well as avoiding the effects between unreliable pairwise views.

---

[4]We borrow the core ideas of information reconstruction applied in vanilla autoencoder (AE [19]), denoising autoencoder (DAE [20]), and masked autoencoder (MAE [21]) and provide three reconstruction regularization options. In a same form, the three kinds of reconstruction loss functions could be formulated as follows:

$$\mathcal{R}_{AE}^v(\mathbf{X}^v, \mathbf{H}^v) = \|\mathbf{X}^v - f_-^v(\mathbf{H}^v; \Omega^v)\|_F^2 = \|\mathbf{X}^v - f_-^v(f^v(\mathbf{X}^v; \Psi^v); \Omega^v)\|_F^2,$$

$$\mathcal{R}_{DAE}^v(\mathbf{X}^v, \tilde{\mathbf{H}}^v) = \left\|\mathbf{X}^v - f_-^v(\tilde{\mathbf{H}}^v; \Omega^v)\right\|_F^2 = \|\mathbf{X}^v - f_-^v(f^v(\mathbf{X}^v + \epsilon; \Psi^v); \Omega^v)\|_F^2, \quad (8)$$

$$\mathcal{R}_{MAE}^v(\mathbf{X}^v, \ddot{\mathbf{H}}^v) = \left\|\mathbf{X}^v - f_-^v(\ddot{\mathbf{H}}^v; \Omega^v)\right\|_F^2 = \|\mathbf{X}^v - f_-^v(f^v(\mathbf{X}^v \odot \mathbf{A}; \Psi^v); \Omega^v)\|_F^2,$$

where $\mathbf{X}^v + \epsilon$ denotes the data disturbed by random Gaussian noise $\epsilon \in \mathbb{R}^{N \times d_v}$ in DAE. $\mathbf{X}^v \odot \mathbf{A}$ is the data masked by random $0 - 1$ matrix $\mathbf{A} \in \{0, 1\}^{N \times d_v}$ in MAE. $\tilde{\mathbf{H}}^v$ and $\ddot{\mathbf{H}}^v$ denote the representations inferred from data $\mathbf{X}^v + \epsilon$ and $\mathbf{X}^v \odot \mathbf{A}$ in DAE and MAE, respectively.

**Theorem 1.** *For any three views ($v \in \{m, n, o\}$), if class mutual information only exists in two views, e.g., $I(\mathbf{y}^m; \mathbf{y}^o) \to 0$, $I(\mathbf{y}^n; \mathbf{y}^o) \to 0$, and $I(\mathbf{y}^m; \mathbf{y}^n) = \delta$, $\delta > 0$, we have minimizing the weighted InfoNCE losses $\mathcal{W}^{m,n} \mathcal{L}_{InfoNCE}^{m,n}(\mathbf{Z}^m, \mathbf{Z}^n) + \mathcal{W}^{m,o} \mathcal{L}_{InfoNCE}^{m,o}(\mathbf{Z}^m, \mathbf{Z}^o) + \mathcal{W}^{n,o} \mathcal{L}_{InfoNCE}^{n,o}(\mathbf{Z}^n, \mathbf{Z}^o)$ is equivalent to maximizing the mutual information between the two views $(e^{\delta/\log N} - 1) I(\mathbf{Z}^m; \mathbf{Z}^n)$.*

Combining with the information losing of each layer through encoder networks, the following theorem further reveals that reconstruction regularization on the hidden features $\mathbf{H}^v$ is conducive to alleviating the losing of discriminative semantic information through data transformation. Hence, we treat the layer output closest to $\mathbf{Z}^v$ in encoders as hidden features to maximize $\prod_{l=t^m+1}^{L^m}(1 - \gamma_l^m)$ and $\prod_{l=t^n+1}^{L^n}(1 - \gamma_l^n)$, aiming at maintaining useful semantic information for contrastive learning.

**Theorem 2.** *For any two views ($v \in \{m, n\}$) with positive class mutual information, denoting $L^v$ as the total layer number of the $v$-th view's encoder network before representation $\mathbf{Z}^v$, the $l$-th layer has the information losing rate $\gamma_l^v \geq 0$. If $\mathbf{S}$ is an oracle variable that contains and only contains multiple views' discriminative semantic information, and $\mathbf{H}^v$ is the $t^v$-th layer's features, we have minimizing the regularized loss $\mathcal{W}^{m,n} \mathcal{L}_{InfoNCE}^{m,n}(\mathbf{Z}^m, \mathbf{Z}^n) + \lambda \sum_v \mathcal{R}^v(\mathbf{X}^v, \mathbf{H}^v)$ is expected to obtain $I(\mathbf{S}; \mathbf{Z}^m; \mathbf{Z}^n) \leq \min\{I(\mathbf{S}; \mathbf{X}^m) \cdot \prod_{l=t^m+1}^{L^m}(1 - \gamma_l^m), I(\mathbf{S}; \mathbf{X}^n) \cdot \prod_{l=t^n+1}^{L^n}(1 - \gamma_l^n)\}$.*

## 4 Experiments

This section validates the effectiveness of our SEM. Specifically, we first conduct comparison experiments on state-of-the-art contrastive learning baselines and SEM with three options of contrastive losses (*i.e.*, $\mathcal{L}_{InfoNCE}, \mathcal{L}_{PSCL}, \mathcal{L}_{RINCE}$). We then conduct ablation studies with three options of weighting strategies (*i.e.*, $\mathcal{W}_{CMI}, \mathcal{W}_{JSD}, \mathcal{W}_{MMD}$), as well as with three options of reconstruction terms (*i.e.*, $\mathcal{R}_{AE}, \mathcal{R}_{DAE}, \mathcal{R}_{MAE}$). Evaluation is built on the concatenation of all views' representations learned by methods. Finally, we show SEM's training process and its hyper-parameter analysis. We provided more experimental results as well as all implementation details of SEM in Appendix.

**Datasets** Our experiments employ five open-source multi-view datasets. Their information is shown in Table 1, where DHA [43] is a depth-included human action dataset where each action has RGB and depth features; CCV [44] refers to the columbia consumer video database whose samples are described with SIFT, STIP, and MFCC features; NUSWIDE [45] collects web images with multiple views (color histogram, block-wise

Table 1: Information of datasets

| Name | View | Size | Class |
|---|---|---|---|
| DHA | 2 | 483 | 23 |
| CCV | 3 | 6,773 | 20 |
| NUSWIDE | 5 | 5,000 | 5 |
| Caltech | 6 | 1,400 | 7 |
| YoutubeVideo | 3 | 101,499 | 31 |

color moments, color correlogram, edge direction histogram, and wavelet texture); Caltech [40] is a widely-used image dataset which leverages six views (Gabor, Wavelet moments, CENTRIST, HOG, GIST, and LBP) to represent samples; YoutubeVideo [46] is a large-scale dataset where each sample has three views including cuboids histogram, HOG, and vision misc. These datasets are diverse in forms and are often organized to comprehensively evaluate the performance of multi-view methods.

### 4.1 Comparison Experiments on Contrastive Learning

**Baselines** K-Means-BSV denotes K-Means clustering results on the best single-view of raw data, and we leverage this baseline to investigate the representation degeneration in comparison methods. InfoNCE [8], PSCL [16], and RINCE [15] are three kinds of CL methods. Since their original versions are designed to handle single views, we extended them to multi-view scenarios as did in [11, 17]. CMC [11], DCP [39], MFLVC [14], and DSIMVC [17] are four kinds of MCL methods. We evaluate our SEM with different contrastive losses (*i.e.*, SEM+InfoNCE, SEM+PSCL, and SEM+RINCE), where the weighting strategy and reconstruction term are fixed to $\mathcal{W}_{CMI}$ and $\mathcal{R}_{AE}$, respectively.

We leverage the linear clustering method K-Means to evaluate the performance of learning representations and report the average results of 10 runs in Table 2. The results indicate that: **I)** Our SEM framework is compatible with different contrastive losses (*e.g.*, InfoNCE, PSCL, and RINCE) and we can clearly observe that SEM+InfoNCE/PSCL/RINCE successfully improve the baselines for large margins. For instance, SEM+InfoNCE respectively outperforms InfoNCE by about 25%, 13%, 4%, 7%, 12% ACC on the five datasets. **II)** MCL approaches could access the semantic information from multiple views, and thus outperforming that from single views. However, a side effect is

Table 2: Linear clustering performance evaluated by ACC and NMI (mean±std%)

| Method | DHA | | CCV | | NUSWIDE | | Caltech | | YoutubeVideo | |
|---|---|---|---|---|---|---|---|---|---|---|
| | ACC | NMI | ACC | NMI | ACC | NMI | ACC | NMI | ACC | NMI |
| K-Means-BSV | 66.6±2.6 | 78.0±1.3 | 19.5±0.3 | 17.8±0.3 | 39.7±0.0 | 11.6±0.0 | 85.1±0.1 | 75.6±0.1 | 16.5±0.6 | 15.9±0.4 |
| InfoNCE [8] | 54.9±3.8 | 77.7±2.1 | 25.8±0.9 | 25.9±0.7 | 56.3±2.2 | 30.3±1.4 | 79.9±1.0 | 71.4±0.9 | 19.6±0.1 | 19.7±0.0 |
| PSCL [16] | 39.8±2.8 | 72.9±2.0 | 21.5±1.6 | 24.3±1.0 | 53.4±0.4 | 27.4±0.5 | 67.8±2.8 | 69.5±2.8 | 15.4±0.3 | 14.3±0.5 |
| RINCE [15] | 49.9±6.2 | 76.3±2.6 | 22.5±0.4 | 23.5±0.2 | 56.6±1.4 | 30.8±1.3 | 80.3±2.2 | 72.0±2.2 | 14.7±0.3 | 13.6±0.2 |
| CMC [11] | 65.0±2.1 | 79.2±1.3 | 21.3±0.4 | 21.8±0.6 | 56.2±1.5 | 24.7±0.9 | 72.7±1.4 | 60.3±1.4 | 19.4±0.3 | 19.6±0.1 |
| DCP [39] | 69.8±2.2 | 82.9±1.6 | 24.1±1.2 | 20.6±0.9 | 48.1±1.4 | 24.5±1.1 | 69.6±6.6 | 66.2±5.3 | 14.0±0.3 | 12.3±0.4 |
| MFLVC [14] | 70.7±1.4 | 81.4±0.8 | 31.6±0.0 | 31.3±0.0 | 55.9±0.0 | 27.4±0.0 | 77.1±0.5 | 67.1±0.6 | 18.3±0.1 | 18.7±0.2 |
| DSIMVC [17] | 63.8±3.0 | 77.2±1.7 | 31.8±0.9 | 30.8±0.6 | 56.7±2.3 | 28.0±1.6 | 76.9±1.7 | 67.3±1.3 | 18.9±0.3 | 18.7±0.2 |
| SEM+InfoNCE | **80.9**±1.9 | **84.1**±0.9 | **39.4**±0.7 | 35.5±0.4 | 60.4±0.4 | 34.9±0.7 | **87.2**±0.3 | **80.3**±0.5 | 31.3±1.1 | 31.1±0.9 |
| SEM+PSCL | 69.7±4.2 | 81.4±1.6 | 39.3±1.1 | **35.9**±0.6 | 57.8±1.4 | 32.6±0.9 | 86.3±1.7 | 78.6±2.1 | **32.2**±0.7 | **32.2**±0.6 |
| SEM+RINCE | 76.3±1.3 | 82.8±0.7 | 38.9±0.9 | 34.6±0.5 | **60.6**±0.7 | **35.6**±1.3 | 85.4±1.4 | 76.7±2.2 | 29.8±0.5 | 29.5±0.5 |

that contrastive learning directly increases the feature representation similarity of multiple views, which might obscure useful discriminative information hidden in high-quality views and lead to the representation degeneration. For example, on DHA and Caltech, results on many MCL methods (*e.g.*, PSCL, RINCE, CMC, and MFLVC) are worse than the single-view baseline K-Means-BSV. **III)** Our SEM not only outperforms all these MCL methods but also mitigates the representation degeneration in MCL, *e.g.*, SEM+PSCL respectively outperforms K-Means-BSV by about 3%, 20%, 18%, 1%, 16% ACC on the five datasets. This is because the framework of SEM is adaptive to multiple views' qualities, which can reduce the side effect between unreliable views with inconsistent information, for better extracting discriminative information and consistent semantics among useful views.

Additionally, we leverage the linear classification method SVM [47] to evaluate the performance of MCL methods to learn representations, where we only use 30% of the learned representations for the training set and the rest for the test set. Figures 3 and 4 show the classification performance on DHA and CCV, respectively. Our SEM improves the baseline methods (especially for PSCL

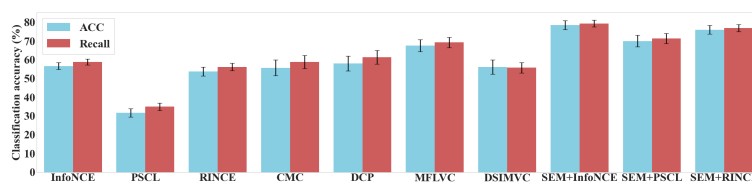

Figure 3: Classification performance on DHA.

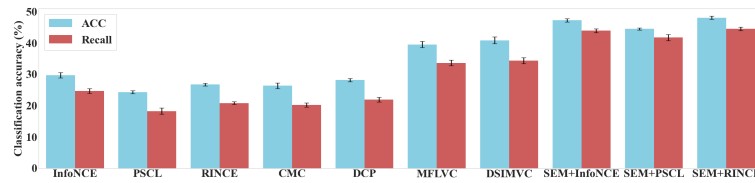

Figure 4: Classification performance on CCV.

and RINCE) and consistently outperforms other MCL methods (such as CMC, DCP, MFLVC, and DSIMVC). Since contrastive learning usually discards the information which is irrelevant to optimization objectives, the results further indicate that the representations learned by MCL are classification-friendly, which generally focus on catching class-level semantics among multiple views.

## 4.2 Ablation Experiments on Self-Weighting Strategy and Reconstruction Regularization

This part presents the ablation experiments to investigate the effectiveness of different weighting strategies $\mathcal{W}_{CMI/JSD/MMD}$ and reconstruction terms $\mathcal{R}_{AE/DAE/MAE}$ in our SEM framework.

Table 3: Clustering accuracy (%) of SEM with different options of weighting strategy $\mathcal{W}$ on two datasets

| | DHA | CCV |
|---|---|---|
| SEM w/o $\mathcal{W}$ | 71.3 | 33.5 |
| SEM w/ $\mathcal{W}_{CMI}$ | 80.9 (↑ 9.6) | 39.4 (↑ 5.9) |
| SEM w/ $\mathcal{W}_{JSD}$ | 80.5 (↑ 9.2) | 35.6 (↑ 2.1) |
| SEM w/ $\mathcal{W}_{MMD}$ | 84.4 (↑ 13.1) | 33.9 (↑ 0.4) |

Table 4: Clustering accuracy (%) of SEM with different options of reconstruction term $\mathcal{R}$ on two datasets

| | DHA | CCV |
|---|---|---|
| SEM w/o $\mathcal{R}$ | 60.5 | 28.7 |
| SEM w/ $\mathcal{R}_{AE}$ | 80.9 (↑ 20.4) | 39.4 (↑ 10.7) |
| SEM w/ $\mathcal{R}_{DAE}$ | 81.5 (↑ 21.0) | 38.4 (↑ 9.7) |
| SEM w/ $\mathcal{R}_{MAE}$ | 83.0 (↑ 22.5) | 39.5 (↑ 10.8) |

Table 3 reports the linear clustering performance (evaluated by ACC) of our SEM framework without self-weighting strategy (*i.e.*, SEM w/o $\mathcal{W}$) and that with three weighting strategies (*i.e.*,

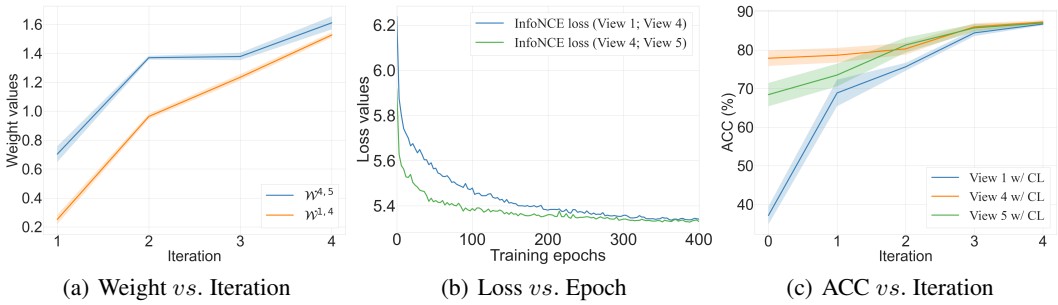

| (a) Weight $vs.$ Iteration | (b) Loss $vs.$ Epoch | (c) ACC $vs.$ Iteration |

Figure 5: (a) The change trend of weights $\mathcal{W}^{1,4}$ and $\mathcal{W}^{4,5}$ in SEM. (b) Loss values $\mathcal{L}^{1,4}_{InfoNCE}$ and $\mathcal{L}^{4,5}_{InfoNCE}$ during contrastive learning. (c) Clustering accuracy on the learned representations of view 1, view 4, and view 5.

$\mathcal{W}_{CMI}, \mathcal{W}_{JSD}, \mathcal{W}_{MMD}$), where the contrastive loss and reconstruction term are fixed to $\mathcal{L}_{InfoNCE}$ and $\mathcal{R}_{AE}$, respectively. Compared with SEM w/o $\mathcal{W}$ (this setting is the reconstruction regularized multi-view contrastive learning) that equally treats contrastive learning between any two views, SEM w/ $\mathcal{W}_{CMI/JSD/MMD}$ can adaptively weight the contrastive learning according to specific two views and thus all these three variants of SEM obtain significant improvements. For example, SEM w/ $\mathcal{W}_{MMD}$ has a 13.1% improvement on DHA and SEM w/ $\mathcal{W}_{CMI}$ has a 5.9% improvement on CCV. Results on more datasets and time costs are shown in Appendix C, where we find that the proposed weighting strategy of class mutual information $\mathcal{W}_{CMI}$ generally achieves the best performance on accuracy and time consumption among the three options of weighting strategy.

Table 4 reports the linear clustering performance (evaluated by ACC) of our SEM framework without reconstruction regularization (*i.e.*, SEM w/o $\mathcal{R}$) and that with three reconstruction terms (*i.e.*, $\mathcal{R}_{AE}, \mathcal{R}_{DAE}, \mathcal{R}_{MAE}$), where the contrastive loss and weighting strategy are fixed to $\mathcal{L}_{InfoNCE}$ and $\mathcal{W}_{CMI}$, respectively. We can easily find that the proposed SEM with reconstruction terms obviously outperforms that without reconstruction terms. For instance, compared with SEM w/o $\mathcal{R}$, SEM w/ $\mathcal{W}_{MAE}$ has 22.5% and 10.8% improvements on DHA and CCV, respectively. This is because the reconstruction regularization makes the hidden features $\{\mathbf{H}^v\}_{v=1}^V$ avoid losing discriminative information, which promotes the multi-view contrastive learning performed on subsequent $\{\mathbf{Z}^v\}_{v=1}^V$. Meanwhile, SEM w/ $\mathcal{R}_{MAE}$ and SEM w/ $\mathcal{R}_{DAE}$ perform better than SEM w/ $\mathcal{R}_{AE}$. This is because, compared with vanilla AE, DAE or MAE (by adding noise or masking on raw data) can make our model more conducive to removing semantic-irrelevant noise as well as capturing hidden patterns.

### 4.3 Experimental Analysis on Mechanism of SEM

This part presents the visualization and analysis on SEM to give an intuition of its behavior and mechanism, where the combination of $\mathcal{L}_{InfoNCE}+\mathcal{W}_{CMI}+\mathcal{R}_{AE}$ is taken as an example.

Let's first recall the views of Caltech dataset in Figure 2(a), we can consider that view 4 and view 5 are high-quality views, while view 1 is a low-quality view. The performance relation among them is $\text{ACC}_{\text{view 4}} > \text{ACC}_{\text{view 5}} > \text{ACC}_{\text{view 1}}$. In Figure 2(c), view 4's representation degeneration occurs.

Figure 5 shows the pairwise weights, losses, and clustering accuracy on Caltech dataset during SEM's training process, where 1 iteration corresponds to 100 epochs, *i.e.*, the step size is set to 100 epochs. Our SEM is a self-weighted multi-view contrastive learning framework that automatically infers different weights for different pairwise views as shown in Figure 5(a), where we can observe that weights $\mathcal{W}^{4,5} > \mathcal{W}^{1,4}$ and they were dynamically updated for 4 times. As a result, contrastive learning between view 4 and view 5 is strengthened by $\mathcal{W}^{4,5}$, while contrastive learning between view 1 and view 4 is weakened by $\mathcal{W}^{1,4}$. Meanwhile, loss $\mathcal{L}^{4,5}_{InfoNCE}$ is minimized earlier than loss $\mathcal{L}^{1,4}_{InfoNCE}$ as shown in Figure 5(b). In other words, since the mutual effect between view 4 and view 5 is strengthened, the effect of view 1 on view 4/view 5 is weakened such that view 4/view 5 does not degenerate. At the same time, the effect of view 4/view 5 on view 1 remains and promotes the representation learning of view 1. Consequently, all views' performance in Figure 5(c) increases through our SEM, and the representation degeneration of view 4 occurring in Figure 2(c) is mitigated.

**Hyper-parameter analysis** Since different datasets have different levels of reconstruction errors, the trade-off coefficient $\lambda$ is introduced to balance the contrastive learning and information recovery in our SEM framework. In Figure 6(a), we change $\lambda$ within the range of $[10^{-3}, 10^{-2}, 10^{-1}, 10^0, 10^1, 10^2, 10^3]$ and report the clustering accuracy tested on representations. The experimental results indicate that SEM is not sensitive to $\lambda$ in $[10^{-1}, 10^1]$. In our experiments, $\lambda$ is consistently set to 1 for all the five datasets. Regarding self-supervised learning, frameworks with fewer manually set hyper-parameters might be more convenient for their applications.

Additionally, we investigate the effect of cluster number when the weight strategy of our SEM framework is selected as $\mathcal{W}_{CMI}$ which needs to pre-define the cluster number when applying K-Means algorithm. As shown in Figure 6(b), when computing the class mutual information, we change the number of clusters within the range of $[K/2, K, 2K, 4K]$ where $K$ denotes the truth class number of multi-view datasets. Compared with $K$, $K/2$ leads to more coarse-grained class mutual information, while $2K$ and $4K$ come in more fine-grained class mutual information. The experimental results demonstrate that SEM with $\mathcal{W}_{CMI}$ is not sensitive to the choices of cluster number.

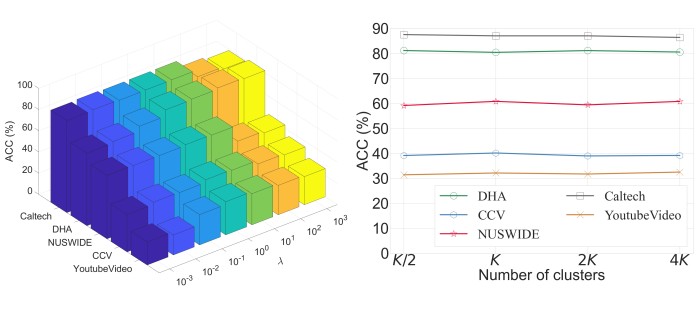

Figure 6: (a) ACC $vs.$ $\lambda$.  (b) ACC $vs.$ $K$.

## 5    Conclusion

In this paper, we showcase that the representation degeneration could seriously limit the application of contrastive learning in multi-view scenarios. To mitigate this issue, we propose self-weighted multi-view contrastive learning with reconstruction regularization (SEM), which is a general framework that is compatible with different options of the contrastive loss, weighting strategy, and reconstruction term. Theoretical and experimental analysis verified the effectiveness of SEM, and it can significantly improve many existing contrastive learning methods in multi-view scenarios. Moreover, ablation studies indicated that SEM is effective with different weighting strategies and reconstruction terms.

Our future work is to extend the proposed SEM to be useful not only for multi-view scenarios, but also for other contrastive learning based domains, such as contrastive learning in sequences. Conceptually, the limitation of the self-weighting strategy is that it is more effective when there are over two views. When there are only two views, the self-weighted multi-view contrastive learning framework transforms into traditional contrastive learning but with reconstruction regularization. Therefore, another future work is to extend the view-level weighting of SEM to sample-level weighting.

## Acknowledgment

This work was supported in part by the National Key Research and Development Program of China under Grant 2022YFA1004100, in part by the Medico-Engineering Cooperation Funds from University of Electronic Science and Technology of China under Grant ZYGX2022YGRH009 and Grant ZYGX2022YGRH014, in part by the National Natural Science Foundation of China under Grant 62276052.

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
