# Appendix for
# Self-Weighted Contrastive Learning among Multiple Views for Mitigating Representation Degeneration

**Jie Xu**[1]    **Shuo Chen**[2]    **Yazhou Ren**[1]    **Xiaoshuang Shi**[1]
**Heng Tao Shen**[1]    **Gang Niu**[2]    **Xiaofeng Zhu**[1,*]
[1]University of Electronic Science and Technology of China, China
[2]RIKEN Center for Advanced Intelligence Project, Japan
*Corresponding Author (seanzhuxf@gmail.com)

We provide supplementary materials for the submission of *Self-Weighted Contrastive Learning among Multiple Views for Mitigating Representation Degeneration*. Specifically, Appendix A (Page-1) shows all theoretical proofs and complexity analysis of SEM; Appendix B (Page-7) includes the settings in experiments; Appendix C (Page-8) lists additional experimental results and provides more experimental analysis, which are not shown in the paper due to space; Appendix D (Page-10) discusses the limitations and future work of this paper. The code implementation, trained models, and datasets used in our method are provided in *https://github.com/SubmissionsIn/SEM*.

## Appendix A    Theoretical Analysis

**Theorem 1.** *For any three views ($v \in \{m, n, o\}$), if class mutual information only exists in two views, e.g., $I(\mathbf{y}^m; \mathbf{y}^o) \to 0$, $I(\mathbf{y}^n; \mathbf{y}^o) \to 0$, and $I(\mathbf{y}^m; \mathbf{y}^n) = \delta$, $\delta > 0$, we have minimizing the weighted InfoNCE losses $\mathcal{W}^{m,n}\mathcal{L}^{m,n}_{InfoNCE}(\mathbf{Z}^m, \mathbf{Z}^n) + \mathcal{W}^{m,o}\mathcal{L}^{m,o}_{InfoNCE}(\mathbf{Z}^m, \mathbf{Z}^o) + \mathcal{W}^{n,o}\mathcal{L}^{n,o}_{InfoNCE}(\mathbf{Z}^n, \mathbf{Z}^o)$ is equivalent to maximizing the mutual information between the two views $(e^{\delta/\log N} - 1)I(\mathbf{Z}^m; \mathbf{Z}^n)$.*

*Proof.* According to Proposition 1, minimizing the weighted InfoNCE losses becomes maximizing the following weighted mutual information:

$$\mathcal{W}^{m,n} I(\mathbf{Z}^m; \mathbf{Z}^n) + \mathcal{W}^{m,o} I(\mathbf{Z}^m; \mathbf{Z}^o) + \mathcal{W}^{n,o} I(\mathbf{Z}^n; \mathbf{Z}^o). \tag{1}$$

Furthermore, based on the definition of CMI weighing strategy, we have

$$\mathcal{W}^{m,n} = e^{\frac{2 \cdot I(\mathbf{y}^m; \mathbf{y}^n)}{H(\mathbf{y}^m) + H(\mathbf{y}^n)}} - 1. \tag{2}$$

If $I(\mathbf{y}^m; \mathbf{y}^o) \to 0$ and $I(\mathbf{y}^n; \mathbf{y}^o) \to 0$, we obtain

$$\lim_{I(\mathbf{y}^m; \mathbf{y}^o) \to 0} \mathcal{W}^{m,o} I(\mathbf{Z}^m; \mathbf{Z}^o) + \lim_{I(\mathbf{y}^n; \mathbf{y}^o) \to 0} \mathcal{W}^{n,o} I(\mathbf{Z}^n; \mathbf{Z}^o)$$
$$= \lim_{\mathcal{W}^{m,o} \to 0} \mathcal{W}^{m,o} \cdot I(\mathbf{Z}^m; \mathbf{Z}^o) + \lim_{\mathcal{W}^{n,o} \to 0} \mathcal{W}^{n,o} \cdot I(\mathbf{Z}^n; \mathbf{Z}^o) = 0. \tag{3}$$

Then, if $I(\mathbf{y}^m; \mathbf{y}^n) = \delta$, $\delta \in \mathbb{R}^+$, Eq. (1) becomes

$$\left( e^{\frac{2 \cdot \delta}{H(\mathbf{y}^m) + H(\mathbf{y}^n)}} - 1 \right) \cdot I(\mathbf{Z}^m; \mathbf{Z}^n). \tag{4}$$

For $H(\mathbf{y}^m) + H(\mathbf{y}^n)$, it has a maximum value $2 \log N$ if $\mathbf{y}^m$ and $\mathbf{y}^n$ follow the uniform distribution, *i.e.*, $H(\mathbf{y}^m) + H(\mathbf{y}^n) = -\sum_{i=1}^{N} p(y_i^m) \log p(y_i^m) - \sum_{i=1}^{N} p(y_i^n) \log p(y_i^n) = -\sum_{i=1}^{N} p(y_i^m) \log \frac{1}{|\mathbf{y}^m|} - \sum_{i=1}^{N} p(y_i^n) \log \frac{1}{|\mathbf{y}^n|} = 2 \log N$. Therefore, we have

$$\left( e^{\frac{2 \cdot \delta}{H(\mathbf{y}^m) + H(\mathbf{y}^n)}} - 1 \right) \cdot I(\mathbf{Z}^m; \mathbf{Z}^n) \geq \left( e^{\delta/\log N} - 1 \right) \cdot I(\mathbf{Z}^m; \mathbf{Z}^n), \tag{5}$$

which completes the proof.    $\square$

37th Conference on Neural Information Processing Systems (NeurIPS 2023).

**Theorem 2.** *For any two views ($v \in \{m, n\}$) with positive class mutual information, denoting $L^v$ as the total layer number of the $v$-th view's encoder network before representation $\mathbf{Z}^v$, the $l$-th layer has the information losing rate $\gamma_l^v \geq 0$. If $\mathbf{S}$ is an oracle variable that contains and only contains multiple views' discriminative semantic information, and $\mathbf{H}^v$ is the $t^v$-th layer's features, we have minimizing the regularized loss $\mathcal{W}^{m,n}\mathcal{L}_{InfoNCE}^{m,n}(\mathbf{Z}^m, \mathbf{Z}^n) + \lambda \sum_v \mathcal{R}^v(\mathbf{X}^v, \mathbf{H}^v)$ is expected to obtain $I(\mathbf{S}; \mathbf{Z}^m; \mathbf{Z}^n) \leq \min\{I(\mathbf{S}; \mathbf{X}^m) \cdot \prod_{l=t^m+1}^{L^m}(1 - \gamma_l^m), I(\mathbf{S}; \mathbf{X}^n) \cdot \prod_{l=t^n+1}^{L^n}(1 - \gamma_l^n)\}$.*

*Proof.* We denote the hidden layers' features in encoders as $\mathbf{H}_{(1)}^v, \mathbf{H}_{(2)}^v, \ldots, \mathbf{H}_{(l)}^v, \ldots, \mathbf{H}_{(L^v)}^v$. Based on data processing inequality, we have

$$I(\mathbf{S}; \mathbf{X}^v) \geq I(\mathbf{S}; \mathbf{H}_{(1)}^v) \geq I(\mathbf{S}; \mathbf{H}_{(2)}^v) \geq \ldots I(\mathbf{S}; \mathbf{H}_{(l)}^v) \geq \ldots I(\mathbf{S}; \mathbf{H}_{(L^v)}^v) \geq I(\mathbf{S}; \mathbf{Z}^v). \quad (6)$$

Considering information losing, we have

$$I(\mathbf{S}; \mathbf{Z}^v) \leq I(\mathbf{S}; \mathbf{X}^v) \cdot \prod_{l=1}^{L^v}(1 - \gamma_l^v). \quad (7)$$

According to Proposition 1 and Proposition 2, minimizing the regularized loss approximately becomes maximizing the following objective:

$$\mathcal{W}^{m,n}I(\mathbf{Z}^m; \mathbf{Z}^n) + \lambda \sum_{v=m,n} I(\mathbf{X}^v; \mathbf{H}^v), \quad (8)$$

where $\mathcal{W}^{m,n} > 0$ as two views ($v \in \{m, n\}$) are with positive class mutual information. The reconstruction regularization $I(\mathbf{X}^v; \mathbf{H}^v)$ makes $I(\mathbf{S}; \mathbf{X}^v) = I(\mathbf{S}; \mathbf{H}^v)$. Therefore, if $\mathbf{H}^v$ is the $t^v$-th layer's features (*i.e.*, $\mathbf{H}_{(t^v)}^v$ act as the regularized hidden features), we have

$$I(\mathbf{S}; \mathbf{Z}^v) \leq I(\mathbf{S}; \mathbf{X}^v) \cdot \prod_{l=t^v+1}^{L^v}(1 - \gamma_l^v). \quad (9)$$

The contrastive loss leads to $\max I(\mathbf{Z}^m; \mathbf{Z}^n)$ which essentially explores the discriminative semantic information in $\mathbf{S}$. Given $I(\mathbf{S}; \mathbf{Z}^m; \mathbf{Z}^n) \leq \min\{I(\mathbf{S}; \mathbf{Z}^m), I(\mathbf{S}; \mathbf{Z}^n)\}$, as a result, we can obtain the mutual information across $\mathbf{S}, \mathbf{Z}^m$, and $\mathbf{Z}^n$ as follows:

$$I(\mathbf{S}; \mathbf{Z}^m; \mathbf{Z}^n) \leq \min\{I(\mathbf{S}; \mathbf{X}^m) \cdot \prod_{l=t^m+1}^{L^m}(1 - \gamma_l^m), I(\mathbf{S}; \mathbf{X}^n) \cdot \prod_{l=t^n+1}^{L^n}(1 - \gamma_l^n)\}. \quad (10)$$

$\square$

In SEM, we have $\mathbf{Z}^v = g^v(\mathbf{H}^v; \Phi^v)$ where $\mathbf{H}^v$ and $\mathbf{Z}^v$ are two different variables. This design aims at separately maintaining different views' discriminative information by $\{\mathbf{H}^v\}_{v=1}^V$ and exploring their common semantic information by $\{\mathbf{Z}^v\}_{v=1}^V$. Contrastive learning on $\{\mathbf{Z}^v\}_{v=1}^V$ will capture the common semantics across multiple views induced by the contrastive loss, and discard other useless information in $\{\mathbf{H}^v\}_{v=1}^V$. In extreme cases, if we consider $g^v$ as a smooth invertible transformation and we have the following theorem:

**Theorem 3.** *For any two views ($v \in \{m, n\}$) with positive class mutual information $I(\mathbf{y}^m; \mathbf{y}^n) = \delta$, $\delta > 0$, if $g^v$ learned by contrastive learning is a smooth invertible transformation, minimizing the regularized loss $\mathcal{W}^{m,n}\mathcal{L}_{InfoNCE}^{m,n}(\mathbf{Z}^m, \mathbf{Z}^n) + \lambda \sum_{v=m,n} \mathcal{R}^v(\mathbf{X}^v, \mathbf{H}^v)$ will lead to a trade-off between $\max I(\mathbf{X}^m; \mathbf{X}^n; \mathbf{H}^m; \mathbf{H}^n)$ and $\max I(\mathbf{X}^m; \mathbf{H}^m) + I(\mathbf{X}^n; \mathbf{H}^n)$.*

*Proof.* According to Proposition 1 and Proposition 2, minimizing the regularized loss approximately becomes maximizing the following objective:

$$(e^{\delta/\log N} - 1)I(\mathbf{Z}^m; \mathbf{Z}^n) + \lambda I(\mathbf{X}^m; \mathbf{H}^m) + \lambda I(\mathbf{X}^n; \mathbf{H}^n). \quad (11)$$

If transformations $g^m$ and $g^n$ are smooth and invertible, the Jacobian determinant is $J_{\mathbf{Z}^m} = |\frac{\partial \mathbf{Z}^m}{\partial \mathbf{H}^m}|$ and $J_{\mathbf{Z}^n} = |\frac{\partial \mathbf{Z}^n}{\partial \mathbf{H}^n}|$, respectively. For the $m$-th and $n$-th views, we have

$$\begin{aligned}
p(\mathbf{h}^m, \mathbf{h}^n) &= p(\mathbf{z}^m, \mathbf{z}^n)J_{\mathbf{Z}^m}(\mathbf{h}^m)J_{\mathbf{Z}^n}(\mathbf{h}^n), \\
p(\mathbf{h}^m) &= p(\mathbf{z}^m)J_{\mathbf{Z}^m}(\mathbf{h}^m), d\mathbf{z}^m = J_{\mathbf{Z}^m}(\mathbf{z}^m)d\mathbf{h}^m, \\
p(\mathbf{h}^n) &= p(\mathbf{z}^n)J_{\mathbf{Z}^n}(\mathbf{h}^n), d\mathbf{z}^n = J_{\mathbf{Z}^n}(\mathbf{z}^n)d\mathbf{h}^n.
\end{aligned} \quad (12)$$

Then, we can obtain the invariance property of mutual information between $I(\mathbf{Z}^m; \mathbf{Z}^n)$ and $I(\mathbf{H}^m; \mathbf{H}^n)$ as follows:

$$
\begin{aligned}
I(\mathbf{Z}^m; \mathbf{Z}^n) &= \int \int p(\mathbf{z}^m, \mathbf{z}^n) \log \left( \frac{p(\mathbf{z}^m, \mathbf{z}^n)}{p(\mathbf{z}^m)p(\mathbf{z}^n)} \right) d\mathbf{z}^m d\mathbf{z}^n \\
&= \int \int \frac{p(\mathbf{h}^m, \mathbf{h}^n)}{J_{\mathbf{Z}^m}(\mathbf{h}^m) J_{\mathbf{Z}^n}(\mathbf{h}^n)} \log \left( \frac{\frac{p(\mathbf{h}^m, \mathbf{h}^n)}{J_{\mathbf{Z}^m}(\mathbf{h}^m) J_{\mathbf{Z}^n}(\mathbf{h}^n)}}{\frac{p(\mathbf{h}^m)p(\mathbf{h}^n)}{J_{\mathbf{Z}^m}(\mathbf{h}^m) J_{\mathbf{Z}^n}(\mathbf{h}^n)}} \right) J_{\mathbf{Z}^m}(\mathbf{z}^m) d\mathbf{h}^m J_{\mathbf{Z}^n}(\mathbf{z}^n) d\mathbf{h}^n \\
&= \int \int p(\mathbf{h}^m, \mathbf{h}^n) \log \left( \frac{p(\mathbf{h}^m, \mathbf{h}^n)}{p(\mathbf{h}^m)p(\mathbf{h}^n)} \right) d\mathbf{h}^m d\mathbf{h}^n \\
&= I(\mathbf{H}^m; \mathbf{H}^n).
\end{aligned}
\tag{13}
$$

As a result, the optimization objective in Eq. (11) becomes

$$
(e^{\delta/\log N} - 1) I(\mathbf{H}^m; \mathbf{H}^n) + \lambda I(\mathbf{X}^m; \mathbf{H}^m) + \lambda I(\mathbf{X}^n; \mathbf{H}^n).
\tag{14}
$$

The mutual information $I(\mathbf{X}^m; \mathbf{X}^n)$ in data $\mathbf{X}^m$ and $\mathbf{X}^n$ is fixed, and the mutual information $I(\mathbf{H}^m; \mathbf{H}^n)$ changes due to variables $\mathbf{H}^m$ and $\mathbf{H}^n$. Maximizing $I(\mathbf{H}^m; \mathbf{H}^n)$ makes variables to access $I(\mathbf{X}^m; \mathbf{X}^n)$, while maximizing $I(\mathbf{X}^m; \mathbf{H}^m) + I(\mathbf{X}^n; \mathbf{H}^n)$ tends to maintain all information of view-specific data in variables. Since $I(\mathbf{X}^m; \mathbf{H}^m) \neq I(\mathbf{X}^m; \mathbf{X}^n; \mathbf{H}^m; \mathbf{H}^n)$, there is a trade-off controlled by $\lambda$, *i.e.*, maximizing to access the mutual information $I(\mathbf{X}^m; \mathbf{X}^n)$ between two view's data, or maximizing $I(\mathbf{X}^m; \mathbf{H}^m) + I(\mathbf{X}^n; \mathbf{H}^n)$ between variables and view-specific data. $\qquad\square$

Typically, $g^v$ will not be a smooth invertible transformation such that $\mathbf{H}^v$ and $\mathbf{Z}^v$ learn different information of data $\mathbf{X}^v$. As we all know, data $\mathbf{X}^v$ in different views usually contain useful discriminative information for common semantics as well as semantic-irrelevant information. We introduce the reconstruction regularization on $\mathbf{H}^v$ to avoid that $\mathbf{H}^v$ loses the useful discriminative information of data (here, $\mathbf{H}^v$ also maintains some semantic-irrelevant information due to the information reconstruction). Then, contrastive learning on $\mathbf{Z}^v$ can make $\mathbf{Z}^v$ access sufficient discriminative information from $\mathbf{H}^v$ to further explore the common semantics of multiple views. However, if the reconstruction regularization is punished on $\mathbf{Z}^v$, $\mathbf{Z}^v$ will also retain the semantic-irrelevant information which might disturb $\mathbf{Z}^v$ to explore the common semantics of multiple views. Therefore, the reconstruction objective of our SEM framework is built on $\mathbf{H}^v$ instead of $\mathbf{Z}^v$, for reducing the interference of semantic-irrelevant information to the contrastive learning performed on $\mathbf{Z}^v$.

**Proposition 1.** *Minimizing the weighted InfoNCE losses among multiple views' representations $\sum_{m,n} \mathcal{W}^{m,n} \mathcal{L}_{InfoNCE}^{m,n}(\mathbf{Z}^m, \mathbf{Z}^n)$ is equivalent to maximizing their weighted mutual information $\sum_{m,n} \mathcal{W}^{m,n} I(\mathbf{Z}^m; \mathbf{Z}^n)$.*

*Proof.* In this part, we leverage $d(\mathbf{z}_i^m, \mathbf{z}_i^n)$ to denote the cosine distance between $\mathbf{z}_i^m \in \mathbf{Z}^m$ and $\mathbf{z}_i^n \in \mathbf{Z}^n$. Then, based on the inequality in Lemma 1, we have:

$$
-\frac{1}{N} \sum_{i=1}^{N} \log \frac{e^{d(\mathbf{z}_i^m, \mathbf{z}_i^n)/\tau}}{\sum_{j=1}^{N} e^{d(\mathbf{z}_i^m, \mathbf{z}_j^n)/\tau}} \geq \log N - I(\mathbf{Z}^m; \mathbf{Z}^n),
\tag{15}
$$

We rewrite the positive and negative pairs in InfoNCE loss and can obtain the following inequality:

$$
\begin{aligned}
&-\frac{1}{N} \sum_{i=1}^{N} \log \frac{e^{d(\mathbf{z}_i^m, \mathbf{z}_i^n)/\tau}}{\sum_{j=1}^{N} \sum_{v=m,n} e^{d(\mathbf{z}_i^m, \mathbf{z}_j^v)/\tau}} \\
&\geq -\frac{1}{N} \sum_{i=1}^{N} \log \frac{e^{d(\mathbf{z}_i^m, \mathbf{z}_i^n)/\tau}}{\sum_{j=1}^{N} e^{d(\mathbf{z}_i^m, \mathbf{z}_j^n)/\tau}} \\
&\geq \log N - I(\mathbf{Z}^m; \mathbf{Z}^n).
\end{aligned}
\tag{16}
$$

Given the equations $I(\mathbf{Z}^m; \mathbf{Z}^n) = I(\mathbf{Z}^n; \mathbf{Z}^m)$ and $\mathcal{W}^{m,n} = \mathcal{W}^{n,m}$, we further have

$$
\begin{aligned}
\sum_{m,n} \mathcal{W}^{m,n} \mathcal{L}_{InfoNCE}^{m,n}(\mathbf{Z}^m, \mathbf{Z}^n) &\geq \sum_{m=1}^{V} \sum_{n=1}^{V} (\log N - \mathcal{W}^{m,n} I(\mathbf{Z}^m; \mathbf{Z}^n)) \\
&= V^2 \log N - 2 \sum_{m=1}^{V} \sum_{n=m}^{V} \mathcal{W}^{m,n} I(\mathbf{Z}^m; \mathbf{Z}^n).
\end{aligned}
\tag{17}
$$

Therefore, $\min \sum_{m,n} \mathcal{W}^{m,n} \mathcal{L}_{InfoNCE}^{m,n}(\mathbf{Z}^m, \mathbf{Z}^n)$ is equivalent to $\max \sum_{m,n} \mathcal{W}^{m,n} I(\mathbf{Z}^m; \mathbf{Z}^n)$, *i.e.*, minimizing the weighted InfoNCE losses among multiple views' representations is equivalent to maximizing their weighted mutual information. $\square$

The success of contrastive learning is often (not absolutely) attributable to the estimation of mutual information. The following Eq. (18) gives the relation between InfoNCE and mutual information, which also has been discussed by other forms in [1, 2, 3, 4, 5]. In this paper, We rewrite a proof to this inequality for the completeness of lemmas.

**Lemma 1.** *Let $m$ and $n$ denote two views, assuming $p(\mathbf{z}_i^m, \mathbf{z}_j^n) = p(\mathbf{z}_i^m)p(\mathbf{z}_j^n)$ when $j \neq i$, we have the following inequality that give the relation between InfoNCE and mutual information:*

$$-\frac{1}{N}\sum_{i=1}^{N}\log\frac{\exp(d(\mathbf{z}_i^m, \mathbf{z}_i^n)/\tau)}{\sum_{j=1}^{N}\exp(d(\mathbf{z}_i^m, \mathbf{z}_j^n)/\tau)} \geq \log N - I(\mathbf{Z}^m; \mathbf{Z}^n). \tag{18}$$

*Proof.* If $j \neq i$, $p(\mathbf{z}_j^n|\mathbf{z}_i^m) = \frac{p(\mathbf{z}_j^n, \mathbf{z}_i^m)}{p(\mathbf{z}_i^m)} = p(\mathbf{z}_j^n)$. Let $\mathcal{S}_i = \sum_{j=1}^{N} \frac{p(\mathbf{z}_i^m, \mathbf{z}_j^n)}{p(\mathbf{z}_i^m)p(\mathbf{z}_j^n)}$, therefore, we have

$$
\begin{aligned}
I(\mathbf{Z}^m; \mathbf{Z}^n) &= \sum_{i=1}^{N}\sum_{j=1}^{N} p(\mathbf{z}_i^m, \mathbf{z}_j^n)\log\frac{p(\mathbf{z}_i^m, \mathbf{z}_j^n)}{p(\mathbf{z}_i^m)p(\mathbf{z}_j^n)} \\
&= \sum_{i=1}^{N}\sum_{j=1}^{N} p(\mathbf{z}_i^m, \mathbf{z}_j^n)\log\left(\frac{p(\mathbf{z}_i^m, \mathbf{z}_j^n)}{p(\mathbf{z}_i^m)p(\mathbf{z}_j^n) \cdot \mathcal{S}_i} \cdot \mathcal{S}_i\right) \\
&= \sum_{i=1}^{N}\sum_{j=1}^{N} p(\mathbf{z}_i^m, \mathbf{z}_j^n)\log\frac{\frac{p(\mathbf{z}_i^m, \mathbf{z}_j^n)}{p(\mathbf{z}_i^m)p(\mathbf{z}_j^n)}}{\mathcal{S}_i} + \sum_{i=1}^{N}\sum_{j=1}^{N} p(\mathbf{z}_i^m, \mathbf{z}_j^n)\log \mathcal{S}_i \\
&= \sum_{i=1}^{N} p(\mathbf{z}_i^m, \mathbf{z}_i^n)\log\frac{\frac{p(\mathbf{z}_i^m, \mathbf{z}_i^n)}{p(\mathbf{z}_i^m)p(\mathbf{z}_i^n)}}{\mathcal{S}_i} + \sum_{i=1}^{N}\sum_{j\neq i} p(\mathbf{z}_i^m, \mathbf{z}_j^n)\log\frac{\frac{p(\mathbf{z}_i^m, \mathbf{z}_j^n)}{p(\mathbf{z}_i^m)p(\mathbf{z}_j^n)}}{\mathcal{S}_i} \\
&\quad + \sum_{i=1}^{N}\sum_{j=1}^{N} p(\mathbf{z}_i^m, \mathbf{z}_j^n)\log \mathcal{S}_i. \\
&= \sum_{i=1}^{N} p(\mathbf{z}_i^m, \mathbf{z}_i^n)\log\frac{\frac{p(\mathbf{z}_i^m, \mathbf{z}_i^n)}{p(\mathbf{z}_i^m)p(\mathbf{z}_i^n)}}{\mathcal{S}_i} + \sum_{i=1}^{N}\sum_{j\neq i} p(\mathbf{z}_i^m, \mathbf{z}_j^n)\log\frac{p(\mathbf{z}_i^m)p(\mathbf{z}_j^n)}{p(\mathbf{z}_i^m)p(\mathbf{z}_j^n)} \\
&\quad + \sum_{i=1}^{N}\sum_{j=1}^{N} p(\mathbf{z}_i^m, \mathbf{z}_j^n)\log \mathcal{S}_i - \sum_{i=1}^{N}\sum_{j\neq i} p(\mathbf{z}_i^m, \mathbf{z}_j^n)\log \mathcal{S}_i \\
&= \sum_{i=1}^{N} p(\mathbf{z}_i^m, \mathbf{z}_i^n)\log\frac{\frac{p(\mathbf{z}_i^m, \mathbf{z}_i^n)}{p(\mathbf{z}_i^m)p(\mathbf{z}_i^n)}}{\mathcal{S}_i} + \sum_{i=1}^{N} p(\mathbf{z}_i^m, \mathbf{z}_i^n)\log \mathcal{S}_i.
\end{aligned}
\tag{19}
$$

Since positive pairs are correlated, we have the estimate: $p(\mathbf{z}_i^m, \mathbf{z}_i^n) \geq p(\mathbf{z}_i^m)p(\mathbf{z}_i^n)$. Therefore, the following inequality holds:

$$
\begin{aligned}
\log \mathcal{S}_i &= \log\left(\sum_{j=1}^{N}\frac{p(\mathbf{z}_i^m, \mathbf{z}_j^n)}{p(\mathbf{z}_i^m)p(\mathbf{z}_j^n)}\right) \\
&= \log\left(\frac{p(\mathbf{z}_i^m, \mathbf{z}_i^n)}{p(\mathbf{z}_i^m)p(\mathbf{z}_i^n)} + \sum_{j\neq i}\frac{p(\mathbf{z}_i^m, \mathbf{z}_j^n)}{p(\mathbf{z}_i^m)p(\mathbf{z}_j^n)}\right) \\
&= \log\left(N + \frac{p(\mathbf{z}_i^m, \mathbf{z}_i^n)}{p(\mathbf{z}_i^m)p(\mathbf{z}_i^n)} - 1\right) \\
&\geq \log N.
\end{aligned}
\tag{20}
$$

According to Lemma 2 and Eq. (20), we assume that there exists a constant $\delta \in (0, 1)$ such that $p(\mathbf{z}_i^m | \mathbf{z}_i^n) \geq \delta, i = 1, 2, \cdots, N$ holds. With the estimation [1, 3], *i.e.*, $p(\mathbf{z}_i^n) \approx \frac{1}{N}, i = 1, 2, \cdots, N$, the following inequality holds:

$$
\begin{aligned}
I(\mathbf{Z}^m; \mathbf{Z}^n) &= \sum_{i=1}^{N} p(\mathbf{z}_i^m, \mathbf{z}_i^n) \log \frac{\frac{p(\mathbf{z}_i^m, \mathbf{z}_i^n)}{p(\mathbf{z}_i^m)p(\mathbf{z}_i^n)}}{\mathcal{S}_i} + \sum_{i=1}^{N} p(\mathbf{z}_i^m, \mathbf{z}_i^n) \log \mathcal{S}_i \\
&\approx \sum_{i=1}^{N} \frac{1}{N} p(\mathbf{z}_i^m | \mathbf{z}_i^n) \log \frac{\frac{p(\mathbf{z}_i^m, \mathbf{z}_i^n)}{p(\mathbf{z}_i^m)p(\mathbf{z}_i^n)}}{\mathcal{S}_i} + \sum_{i=1}^{N} \frac{1}{N} p(\mathbf{z}_i^m | \mathbf{z}_i^n) \log \mathcal{S}_i \\
&\geq \delta \left( \frac{1}{N} \sum_{i=1}^{N} \log \frac{\exp(d(\mathbf{z}_i^m, \mathbf{z}_i^n)/\tau)}{\sum_{j=1}^{N} \exp(d(\mathbf{z}_i^m, \mathbf{z}_j^n)/\tau)} + \log N \right).
\end{aligned}
\tag{21}
$$

Furthermore, we have

$$
-\frac{1}{N} \sum_{i=1}^{N} \log \frac{\exp(d(\mathbf{z}_i^m, \mathbf{z}_i^n)/\tau)}{\sum_{j=1}^{N} \exp(d(\mathbf{z}_i^m, \mathbf{z}_j^n)/\tau)} \geq \log N - \frac{1}{\delta} I(\mathbf{Z}^m; \mathbf{Z}^n).
\tag{22}
$$

Consequently, when the constant $\delta \approx 1$ (*i.e.*, the positive pairs are approximate to be correlated), Eq. (18) holds. □

According to [1], Eq. (22) is more precise when $N$ is larger. Minimizing the left part of Eq. (22) is equivalent to maximizing the mutual information $I(\mathbf{Z}^m; \mathbf{Z}^n)$. Note that this bound is weak as there exists approximation about mutual information [6].

**Lemma 2.** *The optimal value of* $\exp(d(\mathbf{z}_i^m, \mathbf{z}_j^n)/\tau)$ *is proportional to the ratio of* $p(\mathbf{z}_i^m, \mathbf{z}_j^n)$ *to* $p(\mathbf{z}_i^m)p(\mathbf{z}_j^n)$, *i.e.,* $\exp(d(\mathbf{z}_i^m, \mathbf{z}_j^n)/\tau) \propto \frac{p(\mathbf{z}_i^m, \mathbf{z}_j^n)}{p(\mathbf{z}_i^m)p(\mathbf{z}_j^n)}$.

*Proof.* We consider the following formulation:

$$
-\frac{1}{N} \sum_{i=1}^{N} \log \frac{\exp(d(\mathbf{z}_i^m, \mathbf{z}_i^n)/\tau)}{\sum_{j=1}^{N} \exp(d(\mathbf{z}_i^m, \mathbf{z}_j^n)/\tau)}.
\tag{23}
$$

Eq. (23) can be regarded as a cross-entropy loss. As a result, minimizing this loss is equivalent to solving a binary classification problem, namely, classifying the given pairs into positive or negative pairs. We let $\{\mathbf{z}_i^m, \mathbf{z}_i^n\}$ denote the positive pairs and $\{\mathbf{z}_i^m, \mathbf{z}_j^n\}_{j \neq i}$ denote the negative pairs. For each given pairs $\{\mathbf{z}_i^m, \mathbf{z}_j^n\}_{i,j=1}^{N}$, we let $p(\mathbf{z}_i^n | \{\mathbf{z}_1^n, \cdots, \mathbf{z}_N^n\}, \mathbf{z}_i^m)$ denote the predicted probability of finding $\mathbf{z}_i^n$ from $\{\mathbf{z}_1^n, \cdots, \mathbf{z}_N^n\}$ to form positive pairs $\{\mathbf{z}_i^m, \mathbf{z}_i^n\}$. $p(\mathbf{z}_i^m, \mathbf{z}_j^n)$, $p(\mathbf{z}_i^m)$, and $p(\mathbf{z}_j^n)$ denote the joint probability and marginal probabilities of $\mathbf{z}_i^m$ and $\mathbf{z}_j^n$. Then, the optimal value of $p(\mathbf{z}_i^n | \{\mathbf{z}_1^n, \cdots, \mathbf{z}_N^n\}, \mathbf{z}_i^m)$ is:

$$
\begin{aligned}
p(\mathbf{z}_i^n | \{\mathbf{z}_1^n, \cdots, \mathbf{z}_N^n\}, \mathbf{z}_i^m) &= \frac{p(\mathbf{z}_i^n | \mathbf{z}_i^m) \prod_{l \neq i} p(\mathbf{z}_l^n)}{\sum_{j=1}^{N} p(\mathbf{z}_j^n | \mathbf{z}_i^m) \prod_{l \neq j} p(\mathbf{z}_l^n)} \\
&= \frac{\frac{p(\mathbf{z}_i^n | \mathbf{z}_i^m)}{p(\mathbf{z}_i^n)}}{\sum_{j=1}^{N} \frac{p(\mathbf{z}_j^n | \mathbf{z}_i^m)}{p(\mathbf{z}_j^n)}} \\
&= \frac{\frac{p(\mathbf{z}_i^m, \mathbf{z}_i^n)}{p(\mathbf{z}_i^m)p(\mathbf{z}_i^n)}}{\sum_{j=1}^{N} \frac{p(\mathbf{z}_i^m, \mathbf{z}_j^n)}{p(\mathbf{z}_i^m)p(\mathbf{z}_j^n)}}.
\end{aligned}
\tag{24}
$$

The corresponding cross-entropy loss is:

$$
\mathcal{L} = -\frac{1}{N} \sum_{i=1}^{N} \log p(\mathbf{z}_i^n | \{\mathbf{z}_1^n, \cdots, \mathbf{z}_N^n\}, \mathbf{z}_i^m) = -\frac{1}{N} \sum_{i=1}^{N} \log \frac{\frac{p(\mathbf{z}_i^m, \mathbf{z}_i^n)}{p(\mathbf{z}_i^m)p(\mathbf{z}_i^n)}}{\sum_{j=1}^{N} \frac{p(\mathbf{z}_i^m, \mathbf{z}_j^n)}{p(\mathbf{z}_i^m)p(\mathbf{z}_j^n)}}.
\tag{25}
$$

Comparing Eq. (25) with Eq. (23), we can find $\exp(d(\mathbf{z}_i^m, \mathbf{z}_j^n)/\tau) \propto \frac{p(\mathbf{z}_i^m, \mathbf{z}_j^n)}{p(\mathbf{z}_i^m)p(\mathbf{z}_j^n)}$. □

**Proposition 2.** ($\max I(\mathbf{X}^v; \mathbf{H}^v)$ [7]) *Combining with Monte Carlo sampling, minimizing the reconstruction loss between raw data and reconstructed data $\left\| \mathbf{X}^v - f_-^v(\mathbf{H}^v) \right\|_F^2$ is approximate to maximizing the mutual information between raw data and their hidden features $I(\mathbf{X}^v; \mathbf{H}^v)$.*

*Proof.* For the $v$-th view, we let $\mathbf{x}^v$ and $\mathbf{h}^v$ denote the points in the space of raw data $\mathbf{X}^v$ and in the space of hidden features $\mathbf{H}^v$, respectively. According to the definition, the mutual information between $\mathbf{X}^v$ and $\mathbf{H}^v$ can be formulated as

$$I(\mathbf{X}^v; \mathbf{H}^v) = \int_{\mathbf{h}^v} \int_{\mathbf{x}^v} p(\mathbf{x}^v, \mathbf{h}^v) \log \left( \frac{p(\mathbf{x}^v | \mathbf{h}^v)}{p(\mathbf{x}^v)} \right) d\mathbf{x}^v d\mathbf{h}^v. \tag{26}$$

The decoder network achieves the approximation $q(\mathbf{x}^v | \mathbf{h}^v)$ of the true posterior $p(\mathbf{x}^v | \mathbf{h}^v)$. Based on the non-negative property of Kullback-Leibler divergence ($D_{KL}$), we have

$$\int_{\mathbf{x}^v} p(\mathbf{x}^v | \mathbf{h}^v) \log \left( \frac{p(\mathbf{x}^v | \mathbf{h}^v)}{q(\mathbf{x}^v | \mathbf{h}^v)} \right) d\mathbf{x}^v = D_{KL}[p(\mathbf{x}^v | \mathbf{h}^v) || q(\mathbf{x}^v | \mathbf{h}^v)] \geq 0$$

$$\Rightarrow \int_{\mathbf{x}^v} p(\mathbf{x}^v | \mathbf{h}^v) \log \left( p(\mathbf{x}^v | \mathbf{h}^v) \right) d\mathbf{x}^v \geq \int_{\mathbf{x}^v} p(\mathbf{x}^v | \mathbf{h}^v) \log \left( q(\mathbf{x}^v | \mathbf{h}^v) \right) d\mathbf{x}^v$$

$$\Rightarrow \int_{\mathbf{h}^v} p(\mathbf{h}^v) d\mathbf{h}^v \int_{\mathbf{x}^v} p(\mathbf{x}^v | \mathbf{h}^v) \log \left( p(\mathbf{x}^v | \mathbf{h}^v) \right) d\mathbf{x}^v$$

$$\geq \int_{\mathbf{h}^v} p(\mathbf{h}^v) d\mathbf{h}^v \int_{\mathbf{x}^v} p(\mathbf{x}^v | \mathbf{h}^v) \log \left( q(\mathbf{x}^v | \mathbf{h}^v) \right) d\mathbf{x}^v$$

$$\Rightarrow \int_{\mathbf{h}^v} \int_{\mathbf{x}^v} p(\mathbf{x}^v, \mathbf{h}^v) \log \left( p(\mathbf{x}^v | \mathbf{h}^v) \right) d\mathbf{x}^v d\mathbf{h}^v \tag{27}$$

$$\geq \int_{\mathbf{h}^v} \int_{\mathbf{x}^v} p(\mathbf{x}^v, \mathbf{h}^v) \log \left( q(\mathbf{x}^v | \mathbf{h}^v) \right) d\mathbf{x}^v d\mathbf{h}^v$$

$$\Rightarrow \int_{\mathbf{h}^v} \int_{\mathbf{x}^v} p(\mathbf{x}^v, \mathbf{h}^v) \log \left( \frac{p(\mathbf{x}^v | \mathbf{h}^v)}{p(\mathbf{x}^v)} \right) d\mathbf{x}^v d\mathbf{h}^v$$

$$\geq \int_{\mathbf{h}^v} \int_{\mathbf{x}^v} p(\mathbf{x}^v, \mathbf{h}^v) \log \left( \frac{q(\mathbf{x}^v | \mathbf{h}^v)}{p(\mathbf{x}^v)} \right) d\mathbf{x}^v d\mathbf{h}^v$$

$$\Rightarrow I(\mathbf{X}^v; \mathbf{H}^v) \geq \int_{\mathbf{h}^v} \int_{\mathbf{x}^v} p(\mathbf{x}^v, \mathbf{h}^v) \log \left( \frac{q(\mathbf{x}^v | \mathbf{h}^v)}{p(\mathbf{x}^v)} \right) d\mathbf{x}^v d\mathbf{h}^v.$$

Considering $-\int_{\mathbf{h}^v} \int_{\mathbf{x}^v} p(\mathbf{x}^v, \mathbf{h}^v) \log \left( p(\mathbf{x}^v) \right) d\mathbf{x}^v d\mathbf{h}^v \geq 0$, we further have

$$I(\mathbf{X}^v; \mathbf{H}^v) \geq \int_{\mathbf{h}^v} \int_{\mathbf{x}^v} p(\mathbf{x}^v, \mathbf{h}^v) \log \left( q(\mathbf{x}^v | \mathbf{h}^v) \right) d\mathbf{x}^v d\mathbf{h}^v$$

$$- \int_{\mathbf{h}^v} \int_{\mathbf{x}^v} p(\mathbf{x}^v, \mathbf{h}^v) \log \left( p(\mathbf{x}^v) \right) d\mathbf{x}^v d\mathbf{h}^v$$

$$\Rightarrow I(\mathbf{X}^v; \mathbf{H}^v) \geq \int_{\mathbf{h}^v} \int_{\mathbf{x}^v} p(\mathbf{x}^v, \mathbf{h}^v) \log \left( q(\mathbf{x}^v | \mathbf{h}^v) \right) d\mathbf{x}^v d\mathbf{h}^v \tag{28}$$

$$\Rightarrow I(\mathbf{X}^v; \mathbf{H}^v) \geq \int_{\mathbf{x}^v} p(\mathbf{x}^v) d\mathbf{x}^v \int_{\mathbf{h}^v} p(\mathbf{h}^v | \mathbf{x}^v) \log \left( q(\mathbf{x}^v | \mathbf{h}^v) \right) d\mathbf{h}^v.$$

Based on Monte Carlo sampling method [8, 7] on $\mathbf{x}_i^v \in \mathbf{X}^v$, we obtain

$$\int_{\mathbf{x}^v} p(\mathbf{x}^v) d\mathbf{x}^v \int_{\mathbf{h}^v} p(\mathbf{h}^v | \mathbf{x}^v) \log \left( q(\mathbf{x}^v | \mathbf{h}^v) \right) d\mathbf{h}^v$$

$$= \frac{1}{N} \sum_{i=1}^{N} \int_{\mathbf{h}^v} p(\mathbf{h}^v | \mathbf{x}_i^v) \log \left( q(\mathbf{x}_i^v | \mathbf{h}^v) \right) d\mathbf{h}^v \tag{29}$$

$$= \frac{1}{N} \sum_{i=1}^{N} \mathbb{E}_{p(\mathbf{h}^v | \mathbf{x}_i^v)} [\log \left( q(\mathbf{x}_i^v | \mathbf{h}^v) \right)],$$

where $p(\mathbf{h}^v|\mathbf{x}_i^v)$ and $q(\mathbf{x}_i^v|\mathbf{h}^v)$ could be treated as the encoder $f^v$ and decoder $f_-^v$ processes of $\mathbf{x}_i^v$, respectively. There is no harm in supposing that $q(.)$ follows Gaussion distribution [7]. Then, the approximate posterior $q(\mathbf{x}_i^v|\mathbf{h}^v)$ can be formulated as

$$q(\mathbf{x}_i^v|\mathbf{h}^v) = \frac{1}{\sqrt{2\pi}\sigma} \exp\left(-\frac{\left\|\mathbf{x}_i^v - f_-^v(\mathbf{h}^v)\right\|_2^2}{\sigma^2}\right). \tag{30}$$

As a result, we have the following inequality:

$$I(\mathbf{X}^v; \mathbf{H}^v) \geq \frac{1}{N} \sum_{i=1}^N \mathbb{E}_{p(\mathbf{h}^v|\mathbf{x}_i^v)} \left[ -\log\left(\sqrt{2\pi}\sigma\right) - \frac{\left\|\mathbf{x}_i^v - f_-^v(\mathbf{h}^v)\right\|_2^2}{\sigma^2} \right]. \tag{31}$$

Therefore, minimizing $\frac{1}{N} \sum_{i=1}^N \mathbb{E}_{p(\mathbf{h}^v|\mathbf{x}_i^v)} \left[ \left\|\mathbf{x}_i^v - f_-^v(\mathbf{h}^v)\right\|_2^2 \right]$ is approximate to maximizing $I(\mathbf{X}^v; \mathbf{H}^v)$. If we continue to simplify $\mathbf{h}_i^v \in \mathbf{H}^v$ with Monte Carlo sampling method, we further have

$$\frac{1}{N} \sum_{i=1}^N \mathbb{E}_{p(\mathbf{h}^v|\mathbf{x}_i^v)} \left[ \left\|\mathbf{x}_i^v - f_-^v(\mathbf{h}^v)\right\|_2^2 \right] = \frac{1}{N} \sum_{i=1}^N \frac{1}{M} \sum_{j=1}^M \left[ \left\|\mathbf{x}_i^v - f_-^v(\mathbf{h}_{i(j)}^v)\right\|_2^2 \right]. \tag{32}$$

Since $\mathbf{h}_i^v$ can be the only one output of $\mathbf{x}_i^v$ by decoder network [7] (*i.e.*, $M = 1$), we could obtain that minimizing the reconstruction loss $\left\|\mathbf{X}^v - f_-^v(\mathbf{H}^v)\right\|_F^2$ is approximate to maximizing $I(\mathbf{X}^v; \mathbf{H}^v)$. $\square$

**Complexity analysis** Letting $N, n, E$ represent the data size, batch size, and total training epochs, respectively, the computation of loss functions and the update of model parameters are with mini-batch manner. Their time complexity is determined by the batch size $n$ and the total training epochs $E$. Since $n \ll N$ holds, the complexity would be $O(E)$. Letting $V$ represent the number of views, $h_v$ and $z$ denote the dimensionality of $\mathbf{H}^v$ and $\mathbf{Z}^v$ of the $v$-th view, respectively. Step size $S$ denotes the number of training epochs after each update of weights. In terms of weighting strategy $\mathcal{W}_{MMD}$, for reducing the complexity of MMD, we can leverage partial instead of whole samples to update $\{\mathcal{W}^{m,n}\}_{m,n=1}^V$. For example, we can randomly pick up $\hat{n}$ samples ($\hat{n} \ll N$) to compute MMD and the complexity is just $O(\hat{n}^2)$. For weighting strategy $\mathcal{W}_{CMI}$, the computation of $\{\mathcal{W}^{m,n}\}_{m,n=1}^V$ needs $V \times N$ representations from all views to obtain K-Means clustering results and its total time complexity is $O(h_v V N) + O(z V N E/S)$, which is linear to $N$. When $N$ is too large, we can apply mini-batch K-Means to reduce the complexity of CMI weighting strategy.

## Appendix B    Experimental Settings

Table 1: Description for abbreviation

| Abbr. | Description |
|---|---|
| InfoNCE | Info noise contrastive estimation |
| SIFT | Scale-invariant feature transform |
| STIP | Space-time interest points |
| MFCC | Mel-frequency cepstral coefficents |
| CENTRIST | Census transform histogram |
| HOG | Histogram of oriented gradient |
| LBP | Local binary pattern |

The models of all methods are implemented with PyTorch [9] platform and tested on the same device with a NVIDIA GeForce RTX 3090 GPU (24.0GB caches) and a 11th Gen Intel(R) Core(TM) i5-11600KF @ 3.90GHz CPU (64.0GB RAM). For fair comparison, all methods adopt the similar architecture of neural networks following previous work [10, 11]. For our SEM, the encoder network can be denoted as $\mathbf{X}^v \to 500 \to 500 \to 2000 \to \mathbf{H}^v \to \mathbf{Z}^v$ and the decoder is reversed $\mathbf{H}^v \to 2000 \to 500 \to 500 \to \hat{\mathbf{X}}^v$. In this architecture, the penultimate layer of encoder networks is recorded as the hidden features $\mathbf{H}^v$. For all views, the dimension of hidden features $\mathbf{H}^v$ and contrastive representations $\mathbf{Z}^v$ are set to 512 and 128, respectively. Activation function is ReLU [12] and optimizer adopts Adam [13]. For all used datasets, the learning rate is fixed to 0.0003 and the

hyper-parameter $\lambda$ is fixed to 1. Table 2 shows the network training details on different datasets. In our experiments, as computing MMD has high complexity, we select first 2000 samples for avoiding out-of-memory when data size is large. The noise of denoising autoencoder is random Gaussian noise. The mask rate of masked autoencoder is set to 30%. For the CMI weighting strategy, the cluster number of K-Means algorithm is pre-defined to the truth class number of a dataset. For the MMD weighting strategy, the bandwith and number of kernels are set to 4 for all datasets used in this paper. Since our work does not focus on specific contrastive losses, we adopt the fixed parameter settings of InfoNCE/PSCL/RINCE as shown in Table 3 for all experiments. Moreover, the batch size that will affect the number of negative pairs is also fixed to 256.

Table 2: Network training details on different datasets

|  | Pre-training Epoch | CL Epoch | Input dimensions of different views | dimension of $\mathbf{H}^v$ | dimension of $\mathbf{Z}^v$ |
|---|---|---|---|---|---|
| DHA | 100 | 300 | 110/6144 | 512 | 128 |
| CCV | 100 | 200 | 5000/5000/4000 | 512 | 128 |
| NUSWIDE | 100 | 100 | 64/225/144/73/128 | 512 | 128 |
| Caltech | 100 | 400 | 48/40/254/1984/512/928 | 512 | 128 |
| YoutubeVideo | 100 | 25 | 512/647/838 | 512 | 128 |

Table 3: Parameter setting in contrastive losses

|  | Parameters |
|---|---|
| InfoNCE | $\tau = 1.0$ |
| PSCL | $r = 3.0$ |
| RINCE | $\tau = 0.5, \alpha = 0.001, q = 0.5$ |

# Appendix C  Additional Experiments

Figure 1 and 2 show the linear classification performance on NUSWIDE and Caltech datasets. We do not report the results on YoutubeVideo as this large-scale dataset is beyond the usable range of SVM.

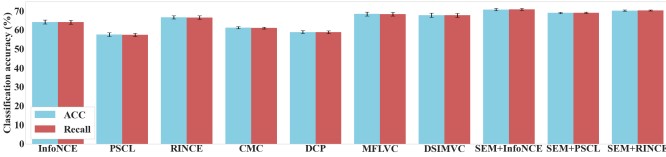

Figure 1: Classification performance on NUSWIDE.

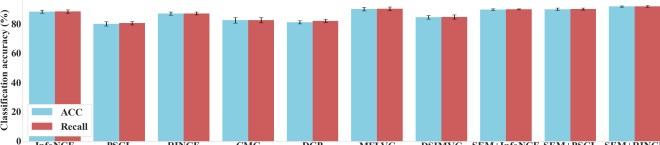

Figure 2: Classification performance on Caltech.

Table 4 reports the time consumption of SEM with three options of weight strategy on five datasets, where the contrastive loss and reconstruction term are fixed to $\mathcal{L}_{InfoNCE}$ and $\mathcal{R}_{AE}$. On CCV, NUSWIDE, and YoutubeVideo, as MMD has high complexity, we select first 2,000 samples to compute weights for avoiding out-of-memory. In this setting, we observe that SEM w/ $\mathcal{W}_{JSD}$ is the fastest variant among the three variants as the computation of JSD is the simplest. Generally, SEM w/ $\mathcal{W}_{CMI}$ is faster than SEM w/ $\mathcal{W}_{MMD}$ even if the MMD is computed on partial data.

Table 5 reports the results of ablation experiments on SEM with different options of weight strategy on five datasets. Table 6 reports the results of ablation experiments on SEM with different options of reconstruction term on five datasets, where $\mathcal{R}_{AE/DAE/MAE}$ w/o SEM denote the performance on representations learned by AE/DAE/MAE models without SEM framework. We can observe that SEM w/ $\mathcal{R}_{AE/DAE/MAE}$ achieve significant improvements over $\mathcal{R}_{AE/DAE/MAE}$ w/o SEM.

Table 4: Time consumption (seconds) of SEM with different options of weight strategy

| Variants | DHA | CCV | NUSWIDE | Caltech | YoutubeVideo |
|---|---|---|---|---|---|
| SEM w/ $\mathcal{W}_{CMI}$ | 38 | 984 | 783 | 533 | 1990 |
| SEM w/ $\mathcal{W}_{JSD}$ | 25 | 556 | 389 | 396 | 1938 |
| SEM w/ $\mathcal{W}_{MMD}$ | 28 | 833 | 1144 | 775 | 2248 |

Table 5: Clustering performance on SEM with different options of $\mathcal{W}$

| Variants | DHA ACC | DHA NMI | CCV ACC | CCV NMI | NUSWIDE ACC | NUSWIDE NMI | Caltech ACC | Caltech NMI | YoutubeVideo ACC | YoutubeVideo NMI |
|---|---|---|---|---|---|---|---|---|---|---|
| SEM w/o $\mathcal{W}$ | 71.29 | 79.77 | 33.50 | 33.01 | 61.90 | 33.82 | 77.71 | 68.68 | 20.96 | 20.82 |
| SEM w/ $\mathcal{W}_{CMI}$ | 80.87 | 84.10 | 39.35 | 35.50 | 60.37 | 34.92 | 87.17 | 80.33 | 31.25 | 31.12 |
| SEM w/ $\mathcal{W}_{JSD}$ | 80.53 | 83.75 | 35.59 | 33.45 | 62.96 | 34.61 | 85.50 | 77.16 | 21.76 | 21.49 |
| SEM w/ $\mathcal{W}_{MMD}$ | 84.40 | 86.22 | 33.89 | 34.13 | 61.39 | 32.75 | 85.67 | 77.36 | 27.50 | 28.47 |

Table 6: Clustering performance on SEM with different options of $\mathcal{R}$

| Variants | DHA ACC | DHA NMI | CCV ACC | CCV NMI | NUSWIDE ACC | NUSWIDE NMI | Caltech ACC | Caltech NMI | YoutubeVideo ACC | YoutubeVideo NMI |
|---|---|---|---|---|---|---|---|---|---|---|
| $\mathcal{R}_{AE}$ w/o SEM | 69.15 | 78.43 | 14.29 | 11.39 | 38.70 | 13.60 | 86.00 | 76.43 | 20.03 | 19.55 |
| $\mathcal{R}_{DAE}$ w/o SEM | 70.39 | 78.87 | 12.67 | 9.58 | 39.54 | 15.08 | 86.43 | 77.47 | 21.73 | 21.49 |
| $\mathcal{R}_{MAE}$ w/o SEM | 69.98 | 77.10 | 14.62 | 11.66 | 35.84 | 14.54 | 86.21 | 77.08 | 22.78 | 21.95 |
| SEM w/o $\mathcal{R}$ | 60.45 | 74.11 | 28.72 | 26.53 | 57.74 | 26.62 | 79.42 | 69.78 | 32.69 | 32.57 |
| SEM w/ $\mathcal{R}_{AE}$ | 80.87 | 84.10 | 39.35 | 35.50 | 60.37 | 34.92 | 87.17 | 80.33 | 31.25 | 31.12 |
| SEM w/ $\mathcal{R}_{DAE}$ | 81.50 | 83.49 | 38.42 | 33.62 | 59.54 | 33.62 | 86.57 | 79.12 | 38.78 | 36.70 |
| SEM w/ $\mathcal{R}_{MAE}$ | 83.02 | 84.44 | 39.48 | 35.79 | 60.94 | 36.24 | 86.71 | 78.03 | 33.26 | 33.04 |

Table 7 reports the experiments on SEM with different sum manner of contrastive losses (where the combination of $\mathcal{L}_{InfoNCE}+\mathcal{W}_{CMI}+\mathcal{R}_{AE}$ is taken). We observe that $\sum_{m}^{V}\sum_{n=m+1}^{V}\mathcal{L}_{CL}^{m,n}$ performs worse than $\sum_{m}^{V}\sum_{n}^{V}\mathcal{L}_{CL}^{m,n}$. This might be because the latter (*i.e.*, $\sum_{m,n}\mathcal{L}_{CL}^{m,n}$ in the paper) pairs negative samples for both $\mathbf{z}_i^m$ and $\mathbf{z}_i^n$ (*e.g.*, $\{\mathbf{z}_i^m,\mathbf{z}_j^v\}_{j\neq i}^{v=m,n}$ and $\{\mathbf{z}_i^n,\mathbf{z}_j^v\}_{j\neq i}^{v=n,m}$), which can access more comprehensive negative sample pairs for contrastive learning than the former.

Table 7: Clustering performance on SEM with different sum manner of contrastive losses

| Variants | DHA ACC | DHA NMI | CCV ACC | CCV NMI | NUSWIDE ACC | NUSWIDE NMI | Caltech ACC | Caltech NMI | YoutubeVideo ACC | YoutubeVideo NMI |
|---|---|---|---|---|---|---|---|---|---|---|
| $\sum_{m}^{V}\sum_{n=m+1}^{V}\mathcal{L}_{CL}^{m,n}$ | 72.04 | 78.38 | 27.95 | 29.92 | 58.28 | 29.74 | 86.54 | 78.35 | 21.47 | 21.82 |
| $\sum_{m}^{V}\sum_{n}^{V}\mathcal{L}_{CL}^{m,n}$ | 80.87 | 84.10 | 39.35 | 35.50 | 60.37 | 34.92 | 87.17 | 80.33 | 31.25 | 31.12 |

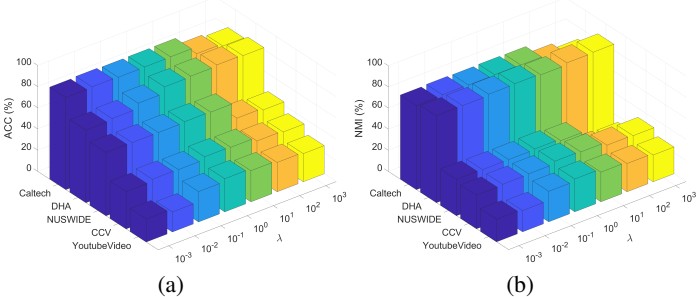

(a)                                        (b)

Figure 3: (a) ACC *vs.* $\lambda$. (b) NMI *vs.* $\lambda$.

**Parameter analysis** Since different datasets have different levels of reconstruction errors, the trade-off coefficient $\lambda$ is introduced to balance contrastive learning and information recovery in our SEM framework. In Figure 3, we change $\lambda$ within the range of $[10^{-3}, 10^{-2}, 10^{-1}, 10^{0}, 10^{1}, 10^{2}, 10^{3}]$

and report the clustering accuracy on the learned representations. The results indicate that SEM framework is not sensitive to $\lambda$ in $[10^{-1}, 10^{1}]$. For all our experiments, $\lambda$ is consistently set to $1$. Additionally, we investigate the effect of cluster number when the weight strategy of SEM framework is selected as $\mathcal{W}_{CMI}$ that needs to pre-define the cluster number when applying K-Means. When computing the class mutual information, as shown in Figure 4, we change the number of clusters within the range of $[K/2, K, 2K, 4K]$ where $K$ denotes the truth class number of datasets. The results demonstrate that SEM with $\mathcal{W}_{CMI}$ is not sensitive to the choices of cluster number.

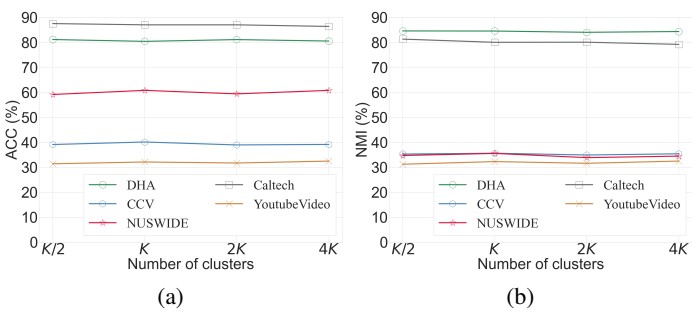

Figure 4: (a) ACC $vs.$ $K$. (b) ACC $vs.$ $K$.

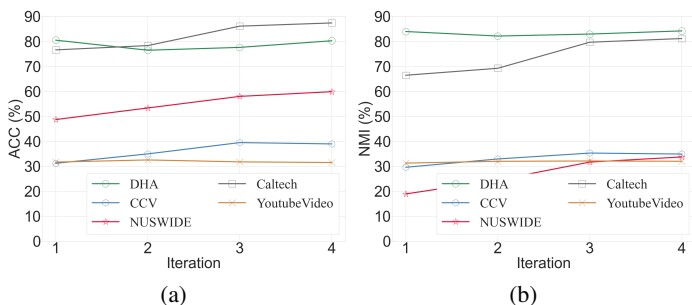

Figure 5: (a) ACC $vs.$ Iterative times of updating weights. (b) NMI $vs.$ Iterative times of updating weights.

In experiments, the times of updating weights during whole training is $E/S$, where $E$ is total training epochs and $S$ is the step size (the number of training epochs after each update of weights). In Figure 5, we fix $S$ after each update of weights and record the clustering accuracy on the learned representations during the iterative times of updating weights in SEM framework (here, an iteration means the one time of updating weights). We observe that only one time of updating weights is enough for some datasets. Usually, the effect of multi-view contrastive learning is gradually improved with the increase of the times of updating weights for some datasets. In our experiments, we fix the times of updating weights on DHA/YoutubeVideo to 1, and fix those on CCV/NUSWIDE/Caltech to 4.

## Appendix D    Social Impacts and Limitations

In multi-view contrastive learning, it might be promising to take the representation degeneration into account, especially in unsupervised environments where the qualities of different views captured by various sensors cannot be guaranteed, *e.g.*, the views from some sensors in real-world application scenarios (such as in animal protection and automatic pilot) are faulty or not applicable, and thus bring semantic-irrelevant information. Additionally, our work proposed a machine learning algorithm to make contrastive learning more practical in the field of multi-view learning. This research is not expected to introduce new negative societal impacts beyond what is already known. Conceptually, the limitation of the self-weighting strategy is that it is more effective when there are over two views. When there are only two views, the self-weighted contrastive learning transforms into traditional contrastive learning but with reconstruction regularization. Therefore, one of our future work is to extend the view-level weighting of our proposed framework to sample-level weighting.