# OpenReview forum: "Self-Weighted Contrastive Learning among Multiple Views for Mitigating Representation Degeneration"
_NeurIPS.cc/2023/Conference — NeurIPS 2023 poster_

### Official Review · Reviewer_tV6r · 2023-06-27

**Soundness:** 4 excellent
**Presentation:** 3 good
**Contribution:** 3 good
**Rating:** 7
**Confidence:** 3

**Summary:**

This paper discusses the limitations of contrastive learning (CL) in multi-view scenarios and proposes a novel framework called SElf-weighted Multi-view contrastive learning (SEM) to address these limitations. The contributions of SEM framework are as follows:
- Alleviating representation degeneration: In multi-view scenarios, CL can lead to representation degeneration when the collected views have inconsistent semantic information or lack sufficient discriminative information. SEM aims to mitigate this issue by adaptively strengthening useful pairwise views and weakening unreliable pairwise views through a self-weighted contrastive loss.
- Regularizing hidden features: SEM introduces a self-supervised reconstruction term to regularize the hidden features of encoders. This regularization assists CL in accessing sufficient discriminative information from the data.
- Extensive experimental validation: Experiments on public multi-view datasets demonstrate that SEM effectively mitigates representation degeneration in existing CL methods and leads to significant performance improvements. Ablation studies further verify the effectiveness of SEM with different options of weighting strategies and reconstruction terms.


**Strengths:**

- The proposal of SEM is well-motivated: SEM tackles the issue of representation degeneration in multi-view scenarios where inconsistent semantic information and insufficient discriminative information exist.
- Flexibility in weighting strategies: SEM provides three options for implementing the weighting strategy, including class mutual information, JS divergence, and maximum mean discrepancy.
- Significant performance improvements and thorough ablation studies: Experimental results on 5 public multi-view datasets, component ablation studies and hyper-parameter analysis demonstrate that SEM effectively mitigates representation degeneration in existing contrastive learning (CL) methods.


**Weaknesses:**

- I have some concerns regarding the rationale behind introducing the reconstruction module. If the aim of incorporating reconstruction is to enhance the discriminative information of the representation, why does the process involve an additional encoder after obtaining the representation H, instead of directly reconstructing the representation z? It would be beneficial to provide more details in the experimental results regarding the position of the reconstruction within the overall encoder and further analyze the relationship between discriminative information and the resulting performance improvement in both the method and experimental sections, thus solidifying the framework.
- (minor) Based on the experimental findings, it appears that the enhancement from the reconstruction regularization is more pronounced compared to the improvement stemming from the self-weighted module. Why the reconstruction leads to a greater improvement needs a more detailed explanation in the experimental section. Besides, in terms of the method's naming, it would be beneficial to highlight the concept of the reconstruction component.
- (minor) The presentation and captions of Figures 1 and 5 can be further improved, and it would be ideal to summarize the conclusions and connections among the data displayed in the three figures.


**Questions:**

This method utilizes a number of models equal to the scale of the views, and it also requires calculating pairwise weights, which significantly increases the memory usage and training time as the number of views increases. The authors seem to lack detailed comparative figures with previous methods in this regard.


**Limitations:**

This paper does not include sufficient discussions about the computational cost and memory overhead, which are suggested to be included.

---

> ### Author Rebuttal · Authors · 2023-08-09
>
> Response To Reviewer tV6r:
>
> >Q1: I have some concerns regarding the rationale behind introducing the reconstruction module. If the aim of incorporating reconstruction is to enhance the discriminative information of the representation, why does the process involve an additional encoder after obtaining the representation $\mathbf{H}^v$, instead of directly reconstructing the representation $\mathbf{Z}^v$?
>
> Thanks for raising this concern. As we all know, data in different views usually contain useful discriminative information for common semantics as well as semantic-irrelevant information. We introduce the reconstruction module on $\mathbf{H}^v$ to avoid that $\mathbf{H}^v$ loses the useful discriminative information of data (here, $\mathbf{H}^v$ also maintains some semantic-irrelevant information due to the information reconstruction). Then, contrastive learning on $\mathbf{Z}^v$ can make $\mathbf{Z}^v$ access sufficient discriminative information from $\mathbf{H}^v$ to explore the common semantics of multiple views. Nevertheless, if the reconstruction module is punished on $\mathbf{Z}^v$, $\mathbf{Z}^v$ also retains the semantic-irrelevant information which might disturb $\mathbf{Z}^v$ to explore the common semantics of multiple views. In this regard, we conduct experiments to investigate the different reconstruction positions and report the clustering accuracy as follows.
>
> &nbsp; |DHA |  CCV | NUSWIDE | Caltech | YoutubeVideo
> ----|----|----|----|----|----
> Reconstruction on $\mathbf{H}^v$  |80.9  |39.4  |60.4  |87.2  |31.3
> Reconstruction on $\mathbf{Z}^v$  |72.4  |27.8  |60.1  |86.6  |30.9
>
> We can observe that the results of the reconstruction on $\mathbf{Z}^v$ have some performance degeneration (especially on DHA and CCV datasets), compared with the results of the reconstruction on $\mathbf{H}^v$. Therefore, the reconstruction of our framework is punished on $\mathbf{H}^v$ instead of $\mathbf{Z}^v$, for reducing the interference of semantic-irrelevant information to the contrastive learning on $\mathbf{Z}^v$.
>
> >Q2: Why the reconstruction leads to a greater improvement needs a more detailed explanation in the experimental section. Besides, in terms of the method's naming, it would be beneficial to highlight the concept of the reconstruction component.
>
> If there are no supervisory signals during data processing, useful discriminative information might lose before contrastive learning. The reconstruction regularization makes the hidden features maintain the discriminative information of data. Then, contrastive learning can access the discriminative information only when they are not lost. Information is well transmitted before it is used and thus the reconstruction leads to great improvements. The self-weighted module and the reconstruction regularization are both important for our method. Hence, we would like to rename our method as "Self-weighted multi-view contrastive learning with reconstruction regularization".
>
>
> >Q3: The presentation and captions of Figures 1 and 5 can be further improved, and it would be ideal to summarize the conclusions and connections among the data displayed in the three figures.
>
> We will improve the presentation and captions of figures. To be specific, the visualization of Figures 2 and 5 are both carried out on the Caltech dataset. Figure 2 shows the representation learning process of traditional contrastive learning, and Figure 5 displays that of our proposed SEM whose framework is illustrated in Figure 1. For example, the useful pairwise views could be view 4 and view 5 and their contrastive learning is strengthened, and the unreliable pairwise views could be view 1 and view 4 and their contrastive learning is weakened.
>
> >Q4: This method utilizes a number of models equal to the scale of the views, and it also requires calculating pairwise weights, which significantly increases the memory usage and training time as the number of views increases. This paper does not include sufficient discussions about the computational cost and memory overhead, which are suggested to be included.
>
> Thanks for this suggestion. To achieve multi-view contrastive learning, previous methods (such as CMC, DCP, MFLVC, and DSIMVC) also need different models to transform different views into the same form, as multi-view data typically involve heterogeneous data forms. Since the mini-batch optimization is adopted, the computational cost of our method (and comparison methods) is linear to the sample size. Additionally, the number of views for multi-view data is often no more than 6 in practical scenarios, which will not increase the unaffordable memory usage and computational burden when calculating pairwise weights. We have put the complexity analysis in Appendix A (Page 7). Additionally, the practical time cost of our method is also shown in Table 3 in Appendix C (Page 8).

---

### Official Review · Reviewer_dnwM · 2023-07-01

**Soundness:** 4 excellent
**Presentation:** 3 good
**Contribution:** 4 excellent
**Rating:** 7
**Confidence:** 5

**Summary:**

This paper researches the representation degeneration of multi-view contrastive learning. To address it, this paper proposes a simple but effective framework of self-weighted multi-view contrastive learning.

**Strengths:**

++The manuscript is well-written and self-consistent. For example, the visualization analysis makes it easy for the reader to understand that considering view differences is necessary for multi-view data.

++The motivation of this work is reasonable, and there are sufficient ablation experiments and theoretical analyses to verify the effectiveness of the proposed SEM algorithm.

++SEM is a general framework that can be adapted to a variety of existing contrastive learning losses, as well as to a variety of autoencoder models.


**Weaknesses:**

--Figure 5(a) shows that weights are updated dynamically in different iterations. Here, are the weights incrementally and linearly increased or are they only updated 4 times?

--The class mutual information weighting strategy is an interesting method that aims to easily measure the discrepancy between views, which compresses the representative class-semantic information of features into one-hot labels. As thus, Eq. (5) seems to be missing some constraints of $\mathbf{Y}^{v*}$ of being the labels.


**Questions:**

In Figure 5(a), are the weights incrementally and linearly increased or are they only updated 4 times?

**Limitations:**

The authors adequately addressed the limitations of this work.

---

> ### Author Rebuttal · Authors · 2023-08-09
>
> Response To Reviewer dnwM:
>
> >Q1: Figure 5(a) shows that weights are updated dynamically in different iterations. Here, are the weights incrementally and linearly increased or are they only updated 4 times?
>
> We are sorry that the illustration of Figure 5(a) is not clear enough. The weights are only updated 4 times during training. The line between the two points just shows their change tendency.
>
> >Q2: The class mutual information weighting strategy is an interesting method that aims to easily measure the discrepancy between views, which compresses the representative class-semantic information of features into one-hot labels. As thus, Eq. (5) seems to be missing some constraints of $\mathbf{Y}^{v*}$ of being the labels.
>
> Thanks for this valuable comment. We will complete the constraints of Eq. (5) that make $\mathbf{Y}^{v*}$ represent one-hot labels, i.e., $s.t. \sum_{j}^{K} y_{ij}^v = 1, y_{ij}^v \in \mathbf{Y}^{v}$.

---

> > ### Comment · Reviewer_dnwM · 2023-08-17
> >
> > Thanks for the efforts in the response. All my concerns have been addressed.

---

### Official Review · Reviewer_USXC · 2023-07-04

**Soundness:** 4 excellent
**Presentation:** 3 good
**Contribution:** 3 good
**Rating:** 7
**Confidence:** 4

**Summary:**

In this paper, the authors show that the representation degradation could limit the application of contrastive learning in multi-view scenarios. To mitigate this issue, they propose the self-weighted multi-view contrastive learning, a general framework that has different options in the contrastive loss, weighting strategy, and reconstruction term. In my opinion, this paper does a good job of showing what is emphasized, with many strengths. I've listed them below, along with some possible weaknesses.

**Strengths:**

#1. The manuscript flows smoothly and is easy to understand. The analysis in the manuscript is well-thought-out and facilitates understanding the motivation and methodology.
#2. Authors show a new multi-view contrastive learning framework that considers handling representation degeneration via self-weighting and information reconstruction.
#3. The framework is technically sound. The manuscript provides three variants of weighting strategy including class mutual information, Jensen-Shannon divergence, and maximum mean discrepancy, and they are with different advantages.
#4. The framework helps existing contrastive learning methods, like InfoNCE, achieve significant performance improvements in multi-view scenarios.
#5. Sufficient ablation experiments and implementation details are provided.

**Weaknesses:**

#1. There is a technical detail that needs to be clarified. SEM adaptively strengthen contrastive learning between the useful pairwise views and also weaken contrastive learning between the unreliable pairwise views. This process can be conducted without supervision by the designed self-weighting framework. But after training, we still don't know which of the representations of multiple views is a good representation. Are the representations learned from useful pairwise views artificially selected to evaluate the performance of downstream clustering tasks?
#2. It would be better for related work to include some recent jobs of contrastive learning.
#3. The abbreviation should be interpreted, such as SIFT, STIP, and MFCC.

**Questions:**

See Weaknesses.

**Limitations:**

The authors have discussed social impacts and limitations.

---

> ### Author Rebuttal · Authors · 2023-08-09
>
> Response To Reviewer USXC:
>
> >Q1: Are the representations learned from useful pairwise views artificially selected to evaluate the performance of downstream clustering tasks?
>
> No. To comprehensively evaluate the performance, our experiments use the concatenation of all learned representations from different views.
>
>
> >Q2: It would be better for related work to include some recent jobs of contrastive learning.
>
> Thanks for this comment. We will further improve the manuscript by discussing more recently published work.
>
> >Q3: The abbreviation should be interpreted, such as SIFT, STIP, and MFCC.
>
> Thanks for the suggestion. We add the following new table to interpret the abbreviations used in this paper.
>
> abbr. | Meaning
> ----|----|
> SIFT | Scale-Invariant Feature Transform
> STIP | Space-Time Interest Points
> MFCC | Mel Frequency Cepstral Coefficents
> CENTRIST | Census transform histogram
> HOG | Histogram of Oriented Gradient
> LBP | Local Binary Pattern

---

> ### Comment · Reviewer_USXC · 2023-08-21
>
> Thanks for the response, I agree with the novelty of this work and keep my accept decision.

---

### Official Review · Reviewer_tvpR · 2023-07-05

**Soundness:** 3 good
**Presentation:** 3 good
**Contribution:** 3 good
**Rating:** 6
**Confidence:** 4

**Summary:**

This work proposed the SEM: SElf-weighted Multi-view contrastive learning framework, which first performs discrepancy measures on representations, and then obtains weights to assist adaptive contrastive learning. Meanwhile, the decoders are leveraged to avoid losing discriminative information. Extensive experiments are conducted to verify the effectiveness of SEM and its different implementation options.

**Strengths:**

In unsupervised environments, it is hard but crucial for multi-view learning to automatically know which view’s features are with useless noise and which view’s features contain useful semantic information. I believe the idea of pair-wise self-weighted contrastive learning is novel. The proposed method obtains the adaptive ability to quality differences among multiple views, and it doesn't require much prior knowledge.

Moreover, multi-view representation learning is important for working with multi-view data as raw views usually have inconsistent semantic meaning when one considers them in practical applications. In this paper, the authors present a robust multi-view contrastive learning method, whose effectiveness is verified by extensive experiments.


**Weaknesses:**

(1) To reduce losing information, the proposed method treats the last layer of encoders as hidden features. It's not clear exactly which layer is as H, and which layer is as Z.

(2) When using the weighting strategies, the framework needs reconstruction pretraining to obtain meaningful representations. The decoder is added to the framework as an auxiliary module. As far as I known, many self-supervised multi-view methods also use autoencoders as the main representation learning module. Some ablation experiments are expected.


**Questions:**

Can the authors show the performance of representation learning relying solely on autoencoders to understand the individual contributions between contrastive learning and reconstruction objectives?

**Limitations:**

This research is not expected to introduce new negative societal impacts.

---

> ### Author Rebuttal · Authors · 2023-08-09
>
> Response To Reviewer tvpR:
>
> >Q1: To reduce losing information, the proposed method treats the last layer of encoders as hidden features. It's not clear exactly which layer is as H, and which layer is as Z.
>
> Thanks for this valuable comment. We present the network setting in Appendix B (Page 7). Specifically, if we think of the network from $\mathbf{X}^v$ to $\mathbf{Z}^v$ as an encoder, then $\mathbf{Z}^v$ is the output of the last layer of the encoder, and $\mathbf{H}^v$ is the output of the penultimate layer of the encoder. We will further clarify this descriptions in the final paper.
>
> >Q2: As far as I known, many self-supervised multi-view methods also use autoencoders as the main representation learning module. Some ablation experiments are expected.
>
> We conduct additional ablation experiments and report the clustering accuracy tested on the learned representations in the following table.
>
> &nbsp; |DHA |  CCV | NUSWIDE | Caltech | YoutubeVideo
> ----|----|----|----|----|----
> vanilla AE|69.2  |14.3  |38.7  |86.0  |20.0
> DCP       |69.8  |24.1  |48.1  |69.6  |14.0
> MFLVC     |70.7  |31.6  |55.9  |77.1  |18.3
> DSIMVC    |63.8  |31.8  |56.7  |76.9  |18.9
> SEM w/ AE |80.9  |39.4  |60.4  |87.2  |31.3
>
> We can observe that, previous AE-based methods (i.e., DCP, MFLVC, and DSIMVC) do not obtain better results than the vanilla AE on some datasets (e.g., Caltech and YoutubeVideo). In comparison, our proposed SEM w/ AE achieves better performance, as it incorporates the self-weighted contrastive learning to handle the representation degeneration.
>
>
> >Q3: Can the authors show the performance of representation learning relying solely on autoencoders to understand the individual contributions between contrastive learning and reconstruction objectives?
>
> Yes. We conduct ablation experiments on three kinds of autoencoders and report clustering accuracy tested on the learned representations as follows.
>
>
> &nbsp; |DHA |  CCV | NUSWIDE | Caltech | YoutubeVideo
> ----|----|----|----|----|----
> AE w/o SEM  |69.2  |14.3  |38.7  |86.0  |20.0
> DAE w/o SEM |70.4  |12.7  |39.5  |86.4  |21.7
> MAE w/o SEM |70.0  |14.6  |35.8  |86.2  |22.8
> SEM w/o AEs  |60.5  |28.7  |57.7	 |79.4  |32.7
> SEM w/ AE   |80.9  |39.4  |60.4  |87.2  |31.3
> SEM w/ DAE  |81.5  |38.4  |59.5  |86.6  |38.8
> SEM w/ MAE  |83.0  |39.5  |60.9  |86.7  |33.3
>
>
> Firstly, the autoencoders have shown their basic representation abilities. They obtain good results on DHA and Caltech but fail on CCV and NUSWIDE (AE/DAE/MAE w/o SEM). Secondly, contrastive learning is conducive to exploring useful mutual information among multiple views. It obtains improvements on CCV and NUSWIDE but fails on DHA and Caltech (SEM w/o AEs). Thirdly, our framework leverages self-weighted contrastive learning while using reconstruction objectives to avoid losing discriminative information. Therefore, SEM with AE/DAE/MAE can obtain significant performance improvements compared with the above results.

---

> > ### Comment · Reviewer_tvpR · 2023-08-14
> >
> > Thanks for providing additional details on the methodology and conducting ablation studies on the autoencoders. Based on the clarification and additional information provided, I will maintain my current rating of acceptance for the paper.

---

### Official Review · Reviewer_gird · 2023-07-05

**Soundness:** 3 good
**Presentation:** 4 excellent
**Contribution:** 3 good
**Rating:** 6
**Confidence:** 5

**Summary:**

In summary, this paper investigates an important question about how to mitigate representation degradation in multi-view contrastive learning. Considering the quality difference of views and losing useful information during nets, the proposed method uses adaptive weighted contrastive learning and adds information reconstruction to improve the performance of contrastive learning in multi-view learning.

**Strengths:**

I) The inductive bias of contrastive learning may allow the newly learned representation to capture trivial information, thus causing representation degradation. So, it is meaningful to propose an effective framework to alleviate this issue in this paper.

II) The authors proposed a novel multi-view contrastive learning framework called SEM with multiple implementations. For example, the paper provides different weighting strategies with different advantages and experiments show their effectiveness.

III) Comparison experiments on different datasets show that SEM+infoNCE,RINCE,PSCL has a significant improvement over infoNCE,RINCE,PSCL themself, indicating the effectiveness.

IV) The supplementary material is well organized, including detailed appendices as well as code.


**Weaknesses:**

I) Due to the high complexity of MMD computation, it seems difficult to obtain MMD results on YoutubeVideo dataset (over 100,000 samples).

II) Introducing weighting strategies may increase the number of hyper-parameters. It might be better to add more descriptions, e.g., MMD.

III) Some grammatical mistakes need to be corrected, e.g., transfers is transfer in Line 155; adaptively weighting is adaptively weight in Line 304.


**Questions:**

Please see weaknesses.

**Limitations:**

The authors addressed the limitations and potential negative societal impact of their work.

---

> ### Author Rebuttal · Authors · 2023-08-09
>
> Response To Reviewer gird:
>
> >Q1: Due to the high complexity of MMD computation, it seems difficult to obtain MMD results on YoutubeVideo dataset (over 100,000 samples).
>
> Yes, it is indeed difficult to obtain weights of the MMD weighting strategy on YoutubeVideo. Therefore, for reducing the computation complexity of MMD, we leveraged partial instead of whole samples when applying the MMD weighting strategy. We already discussed this in complexity analysis in Appendix A (Page 7). Furthermore, Appendix B (Page 7) provides the implementation details when applying the MMD weighting strategy. That is, the weights of the MMD weighting strategy are computed by only leveraging the first 2,000 samples on YoutubeVideo, and thus the computation complexity of our method is controllable in practice uses.
>
> >Q2: Introducing weighting strategies may increase the number of hyper-parameters. It might be better to add more descriptions, e.g., MMD.
>
> In our experiments, for the CMI weighting strategy, the cluster number is pre-defined to the truth class number of a dataset. For the MMD weighting strategy, the bandwith and number of kernels are set to 4 on all datasets used in this paper.
>
> >Q3: Some grammatical mistakes need to be corrected, e.g., transfers is transfer in Line 155; adaptively weighting is adaptively weight in Line 304.
>
> Thanks for these comments. We will correct the grammatical mistakes and further polish our draft in the final version.

---

### Comment · Area_Chair_fCZT · 2023-08-16
**Discussion**

Dear Reviewers,

I appreciate your efforts thus far. Please read the author's rebuttals and other reviews attentively and respond to at least acknowledge that you've seen them. If your evaluation of the paper changes, please update your score and briefly explain the difference.

The paper received diverging initial reviews. Please consider discussing whether we can reach a consensus with the authors or other reviewers.

Thank you,
AC

---

### Comment · Area_Chair_fCZT · 2023-08-19
**Look forward to further feedback**

Dear Reviewers,

The open discussion phase of the paper is nearing its end, and the authors have provided more detailed elaboration, ablation studies, and explanations of the principles in the rebuttal phase, mainly in response to the proposed algorithms. In order to ensure the smooth running of the conference, we would like to receive your responses to the authors' rebuttals as soon as possible. Therefore, we kindly ask you to submit your feedback as early as possible, if possible. Once again, we thank you for your time and look forward to your valuable comments.

Thank you,
AC

---

### Decision · Program_Chairs · 2023-09-21

**Decision:**

Accept (poster)

**Comment:**

This paper proposes a self-weighted multi-view contrastive learning framework that enables the model to adaptively enhance good pair-wise view representations by measuring the difference between representations. The results show that this method effectively alleviates the representation degradation problem of contrastive learning and significantly improves the performance of contrastive learning on public multi-view datasets. The paper is clearly written and well-analyzed. The novelty of this method mainly lies in the adaptive multi-view contrastive learning framework. The author's response successfully addresses some of the reviewers' concerns and provides additional (supportive) experimental results. The detailed settings of the experiment and grammatical errors need to be improved, and the related work of the paper can be further supplemented. Authors are encouraged to consider the reviewers' detailed comments in the final version. For the above reasons, the AC suggested accepting this paper.